EMBO
*reports*

# The CDK Pho85 inhibits Whi7 Start repressor to promote cell cycle entry in budding yeast

Cristina Ros-Carrero ⓘ, Mihai Spiridon-Bodi ⓘ, J Carlos Igual ⓘ ✉ & Mercè Gomar-Alba ⓘ ✉

## Abstract

Pho85 is a multifunctional CDK that signals to the cell when environmental conditions are favorable. It has been connected to cell cycle control, mainly in Start where it promotes the G1/S transition. Here we describe that the Start repressor Whi7 is a key target of Pho85 in the regulation of cell cycle entry. The phosphorylation of Whi7 by Pho85 inhibits the repressor and explains most of the contribution of the CDK in the activation of Start. Mechanistically, Pho85 downregulates Whi7 protein levels through the control of Whi7 protein stability and *WHI7* gene transcription. Whi7 phosphorylation by Pho85 also restrains the intrinsic ability of Whi7 to associate with promoters. Furthermore, although Whi5 is the main Start repressor in normal cycling cells, in the absence of Pho85, Whi7 becomes the major repressor leading to G1 arrest. Overall, our results reveal a novel mechanism by which Pho85 promotes Start through the regulation of the Whi7 repressor at multiple levels, which may confer to Whi7 a functional specialization to connect the response to adverse conditions with the cell cycle control.

**Keywords** Start; Cell Cycle; Pho85; Whi7; Whi5
**Subject Category** Cell Cycle

## Introduction

The cyclin-dependent kinase (CDK) family encompasses a series of serine-threonine protein kinases, many of which command the central mechanisms of the cell cycle engine. Their differential activity depends on the formation of a heterodimeric complex with the corresponding activating subunit, the cyclin. There are multiple CDKs in eukaryotes, with up to 20 in human cells (Malumbres, 2014). The founding member of this family, CDK1, was discovered in the 1980s in yeast and Xenopus oocytes, placing CDK activity as the master regulator of the cell cycle (Russell and Nurse, 1986; Maller, 1998). Since these initial studies, the importance of CDKs arose not only in controlling cell division but also in other fundamental cellular processes, such as transcription, in response to external and internal cues.

The budding yeast *S. cerevisiae* has six CDKs: Cdc28, Pho85, Kin28, Srb10, Bur1, and Ctk1. Among these, only Cdc28 (CDK1 in mammals) is essential for cell proliferation. Cdc28 is associated with nine different cyclins that regulate the different cell cycle transitions. Thus, it controls the G1/S transition through its association with G1 cyclins Cln1-3, the S phase through its association with S-phase cyclins Clb5-6, and mitosis through its association with G2/M cyclins Clb1-4 (Enserink and Kolodner, 2010; Mendenhall and Hodge, 1998).

Relevant to this work is the control of Start in the G1/S transition. Start is a critical point in the eukaryotic cell cycle in which cells irreversibly commit to initiate a new round of division. Start execution involves the activation of a specific transcriptional program that induces the sharp expression of many genes coding for cell cycle regulators and proteins involved in cellular events that occur at this early stage of the cell cycle (Johnson and Skotheim, 2013). In yeast, this program is mediated by the transcription factors SBF (Swi4-Swi6) and MBF (Mbp1-Swi6) (Haase and Wittenberg, 2014). The importance of the Start transcriptional program is evident from the lethality of *swi4swi6* and *swi4mbp1* double mutants (Koch et al, 1993). Although SBF is bound to the G1/S gene promoters in early G1, Start transcription is blocked because of the association of transcriptional repressors Whi5 and Whi7, which are the functional homologs of Retinoblastoma (Rb) in mammalian cells (Gomar-Alba et al, 2017; de Bruin et al, 2004; Costanzo et al, 2004). Start gene expression is initially triggered by Cdc28-Cln3 kinase in late G1, which promotes basal transcription of SBF-dependent genes (Kõivomägi et al, 2021). As a result, Cdc28-Cln1 and Cdc28-Cln2 accumulate and phosphorylate the Whi5 and Whi7 repressors as well as the SBF and MBF transcriptional factors to fully activate the Start transcriptional program (Johnson and Skotheim, 2013), establishing a positive feedback loop that gives coherence to the G1/S transition and drives cellular irreversible commitment to S phase entry (Skotheim et al, 2008; Charvin et al, 2010).

Whi5 and Whi7 show sequence homology and similar cell cycle-regulated localization; however, functional distinctions between both repressors have been described. Specifically, they have different determinants for binding to promoters and show different preferences for Start promoter genes (Méndez et al, 2020). In addition, Whi7 negatively regulates Start through the retention of the Cln3 cyclin in the endoplasmic reticulum membrane (Yahya et al, 2014). Whi7 is a less abundant protein and shows weaker nuclear accumulation than Whi5, which could explain why Whi5 is

Institut de Biotecnologia i Biomedicina (BIOTECMED) and Departament de Bioquímica i Biologia Molecular, Universitat de València, 46100 Burjassot, Spain.
✉E-mail: jcigual@uv.es; merce.gomar@uv.es

the main transcriptional repressor under normal conditions (Gomar-Alba et al, 2017; Méndez et al, 2020). Recent studies have suggested that Whi5 and Whi7 may be differentially regulated under adverse conditions. Whi5 nuclear amount increases during some environmental stresses (Qu et al, 2019), and upon nutrient removal, Whi5 relocates to the nucleus in the S phase (Irvali et al, 2023). Furthermore, *WHI7* expression, but not *WHI5*, is significantly upregulated under various stress conditions (Waern and Snyder, 2013; Gasch et al, 2000; García et al, 2004). Besides, *WHI7* expression is completely dependent on the Cell Wall Integrity (CWI) pathway under both normal and cell wall stress conditions (Méndez et al, 2020; Sanz et al, 2018), suggesting that Whi7 may be an important mediator of CWI pathway functions connecting cell cycle regulation with the cellular response to cell wall damage. In fact, under cell wall stress, Whi7, and not Whi5, becomes the main Start repressor leading to G1 arrest (Méndez et al, 2020). Finally, a new Whi7-specific role in quiescence has been proposed (Miles and Breeden, 2022).

In addition to Cdc28, the CDK Pho85 (Cdk5 in mammals) has been linked to the control of the cell cycle, particularly to Start control. Pho85 was originally identified as a pivotal regulator of cellular response to phosphate (Lenburg and O'Shea, 1996). Pho85, associated with Pho80 cyclin, inhibits the transcription factor Pho4 under optimal phosphate conditions; however, under phosphate deficiency, Pho85 is inhibited, which causes the activation of Pho4, which in turn triggers a transcriptional response of the *PHO* genes involved in the cellular response to phosphate deprivation (Uesono et al, 1987; Kaffman et al, 1994; O'Neill et al, 1996). Later, Pho85 has been related to various cellular functions, constituting a clear example of a multitasking CDK (reviewed in Huang et al, 2007; Carroll and O'Shea, 2002). This pleiotropic role explains why *pho85* deletion results in a wide spectrum of defects such as slow growth in rich media, increased G1 duration, lethality in non-fermentable carbon sources, accumulation of glycogen, polarity defects and hypersensitivity to cell wall stress, osmotic stress, high concentrations of calcium or sodium, and DNA damage. Therefore, Pho85 is at the center of the metabolic response to many unfavorable environmental conditions. The picture that emerges is that Pho85 is generally active when environmental conditions are satisfactory and negatively regulates the expression of genes involved in the cellular response to various stresses (Carroll et al, 2001; Nishizawa et al, 2004) through the inhibition of transcription factors such as Pho4, Crz1 or Gcn4 (Sopko et al, 2006; Bömeke et al, 2006; Shemer et al, 2002).

Up to ten Pho85-activating cyclins have been identified, which confer to the CDK substrate specificity and differential regulation depending on the particular signals (Carroll and O'Shea, 2002; Measday et al, 1997). Pho85 cyclins are classified into two subfamilies based on their sequence in the cyclin box and their functions. On the one hand, the Pho80 subfamily encompasses the Pho80, Pcl6, Pcl7, Pcl8 and Pcl10 cyclins. When associated with these cyclins, Pho85 is mainly involved in the regulation of metabolism and response to environmental changes. On the other hand, Pho85 can also associate with the cyclins of the Pcl1,2 subfamily: Pcl1, Pcl2, Pcl9, Clg1 and Pcl5. These Pho85-cyclins complexes are mainly involved in the control of cell cycle progression. In fact, *PCL1*, *PCL2*, and *PCL9* genes are periodically expressed, peaking at the M/G1 or G1/S boundaries (Measday et al, 1997; Tennyson et al, 1998; Aerne et al, 1998). Despite this traditional functional distinction, the functions of these two cyclin

subfamilies are intertwined. For example, Pho80 is involved in cell cycle control under physiological conditions and in response to phosphate stress or DNA damage (Menoyo et al, 2013; Nishizawa et al, 1998; Wysocki et al, 2006), and has also been linked to cell cycle arrest during the transition to G0 in the stationary phase (Wanke et al, 2005). In addition, Clg1 and Pcl2 control cell cycle progression in response to nitrogen limitation (Truman et al, 2012); Pcl5 acts in nutrient sensing (Aviram et al, 2008); and Pcl7, which regulates glycogen metabolism, is periodically expressed (Lee et al, 2000). These interconnections point to Pho85 as a key CDK to mediate cell cycle control in response to nutrient availability and stress.

As mentioned, the *pho85* mutant shows a delayed G1 phase (Wysocki et al, 2006) and the Pho85 cyclins with periodic expression peaks around G1, suggesting that the role of Pho85 in cell cycle control occurs mainly in Start. Reinforcing this idea, Pho85 is essential for cell cycle progression in the absence of Cdc28 G1 cyclins (Measday et al, 1994; Espinoza et al, 1994). Indeed, evidence has accumulated directly connecting Pho85 to key G1 regulators in order to promote Start (Jiménez et al, 2013). Pho85-Pho80 stabilizes the Cln3 cyclin by phosphorylating Ser449 and Thr520, and upon phosphate deficiency the inactivation of Pho85 causes Cln3 unstabilization, which contributes to the G1 transient arrest observed in this condition (Menoyo et al, 2013). Moreover, Pho85-Pcl9 phosphorylates the transcriptional repressor Whi5, which collaborates with Cdc28-Cln in the release of histone deacetylases Rpd3 and Hos3 from Start promoters, leading to induction of gene expression (Huang et al, 2009). Finally, Pho85-Pcl1 phosphorylates the CDK-inhibitor Sic1 targeting it for degradation (Nishizawa et al, 1998), a mechanism that is involved in cell cycle re-entry after DNA damage (Wysocki et al, 2006).

In this manuscript, we describe a new key target of the CDK Pho85 in Start. Specifically, Pho85 facilitates cell cycle progression by restricting the activity of the Start repressor Whi7 at various levels. As a consequence, when Pho85 is inactivated, both Whi7 cellular levels and Whi7 intrinsic ability to associate with Start promoters increase, becoming Whi7 the most important Start transcriptional repressor to stop cell cycle progression.

# Results

## Whi7 Start transcriptional repressor is an hyperphosphorylated protein

The CDK Cdc28/CDK1 is the main kinase known to be involved in the cell cycle regulation of the Start transcriptional repressor Whi7 (Yahya et al, 2014; Gomar-Alba et al, 2017). The CDK Pho85/CDK5 is a multifunctional kinase that has been linked to stress signaling and Start control through different pathways and molecular effectors (Jiménez et al, 2013; Huang et al, 2007; Carroll and O'Shea, 2002). Because Whi7 is a Start repressor with functional specialization during stress (Méndez et al, 2020), we asked whether Whi7 could be a target of CDK Pho85. To investigate this possibility, we first applied phosphoproteomics of Whi7 in wild-type and *pho85* cells. We carried out TiO2 phosphopeptide enrichment and tandem mass spectrometry (LC-MS/MS) using purified Whi7-GFP mildly induced with the β-estradiol-*GAL* system (Louvion et al, 1993). The analyses of the

**Table 1. Whi7 phosphorylation sites in wild-type and *pho85* mutant cells.**

| wt | pho85 |
|---|---|
| **T5** | |
| **T14** | **T14** |
| **S27** | **S27** |
| **T32** | |
| **S34** | |
| **S36** | **S36** |
| **T40** | **T40** |
| **S56** | |
| **S62** | **S62** |
| Y71 | |
| Y73 | Y73 |
| **T84** | |
| **S98** | **S98** |
| **T100** | **T100** |
| **S105** | **S105** |
| T151 | T151 |
| T158 | S158 |
| S172 | S172 |
| S186 | |
| **S212** | **S212** |
| T222 | T222 |
| S224 | S224 |
| T225 | T225 |
| S229 | S229 |

Three independent biological replicates of each strain were analyzed. The indicated phosphosites were detected in at least 2 of the 3 biological replicates. CDK consensus phosphorylation sites are shown in bold.

phosphopeptides revealed the existence of 24 phosphorylation sites in wild-type cells (Table 1). They include 11 out of the 12 CDK consensus sites (Cdc28 or Pho85). This result demonstrate that Whi7 exist as a highly phosphorylated protein, a common trait of the Start transcriptional repressors like yeast Whi5 and mammalian Rb proteins (Wagner et al, 2009; Hasan et al, 2013). As for the *pho85* mutant cells, 17 phosphorylated sites (including 8 CDK consensus sites) were identified, indicating that Whi7 remains as a hyperphosphorylated protein in *pho85* mutant cells (Table 1). In addition to the loss of three CDK sites, we found that several of the identified CDK consensus sites showed a reduced relative phosphorylation compared to non-CDK sites in the *pho85* mutant (Appendix Fig. S1 and Appendix Table S1). We conclude that Whi7 is an hyperphosphorylated protein and may be regulated by multiple protein kinases in addition to the CDK Cdc28, including the CDK Pho85.

## The CDK Pho85 downregulates the cellular levels of Whi7

To investigate the possible connection between the Start repressor Whi7 and the CDK Pho85, we next analyzed Whi7

phosphorylation and protein levels by western blotting. No significant changes in the phosphorylation pattern could be observed in the *pho85* mutant, probably because CDKs Cdc28 and Pho85 could be redundant in the phosphorylation of Whi7 in multiple CDK sites, as the phosphoproteomic analysis suggest. However, we found that the cellular levels of Whi7 protein were increased 3–4 times upon inactivation of Pho85 (Fig. 1A), being the upregulation of Whi7 in *pho85* abolished by the addition of a plasmid expressing the *PHO85* gene (Fig. 1B). Then, we analyzed *WHI7* gene expression in the *pho85* mutant strain. The results indicated a twofold increase in *WHI7* mRNA levels (Fig. 1C). Whi7 is a highly unstable protein (Gomar-Alba et al, 2017). Therefore, we finally determined the kinetics of Whi7 degradation in the *pho85* mutant using translational shut-offs, and we calculated the degradation rate constant ($k_d$) from the slope of the linear regression obtained from the quantification of the western blots. This assay revealed that the inactivation of *pho85* increased Whi7 protein stability (Fig. 1D). Collectively, these results demonstrate that Whi7 is negatively regulated by Pho85 at two different levels: gene expression and protein stability. Importantly, this is a Whi7-specific mechanism, as protein levels of the other Start transcriptional repressor, Whi5, are barely affected by Pho85 activity (Fig. 1E).

The subcellular localization of Whi7 and Whi5 is cell cycle regulated, accumulating in the nucleus from mitotic exit to Start (Méndez et al, 2020). We wondered whether Pho85 could control the subcellular localization of both repressors. However, the localization of Whi7-NeonGreen and Whi5-NeonGreen was roughly similar in *pho85* when compared to the wild-type strain (Fig. EV1A). To further investigate Whi7 subcellular localization in *pho85*, we used Htb2-mCherry as nuclear marker and we slightly induced Whi7 protein levels with the β-estradiol-*GAL* system by incubating the cells with β-estradiol 1 nM for 1 h. (Fig. EV1B). We observed that the absence of Pho85 does not affect the cell cycle regulated subcellular localization of Whi7 or the percentage of Whi7 protein amount accumulated in the nucleus of G1 cells (Fig. EV1C,D).

## The CDK Pho85 acts with multiple cyclins to control Whi7

We next sought to identify the specific cyclins that collaborate with Pho85 in the regulation of Whi7. The CDK Pho85 associates with multiple cyclins, grouped in the Pho80 subfamily cyclins, which are mostly involved in phosphate metabolism and the response to changes in the external conditions, and the Pcl1,2 subfamily cyclins, more related to cell cycle regulation (Jiménez et al, 2013).

In order to determine which cyclins are involved in the downregulation of Whi7 by Pho85, we measured Whi7 levels in distinct cyclin mutant strains. As shown in Fig. 2A, the inactivation of Pho80 and the combination of Pcl1, Pcl2, Pcl9 and Pho80 mutations caused an increase in Whi7 protein levels, suggesting that these cyclins are involved in the regulation of Whi7. Next, we investigated the role of the different cyclins in the two regulatory mechanisms described above. Whereas inactivation of Pcls cyclins did not affect *WHI7* mRNA levels, *pho80* mutant cells showed increased levels of *WHI7* mRNA, resembling the result observed in the *pho85* mutant strain (Fig. 2B). In contrast, analysis of Whi7 protein stability showed that the cyclins from both families play a

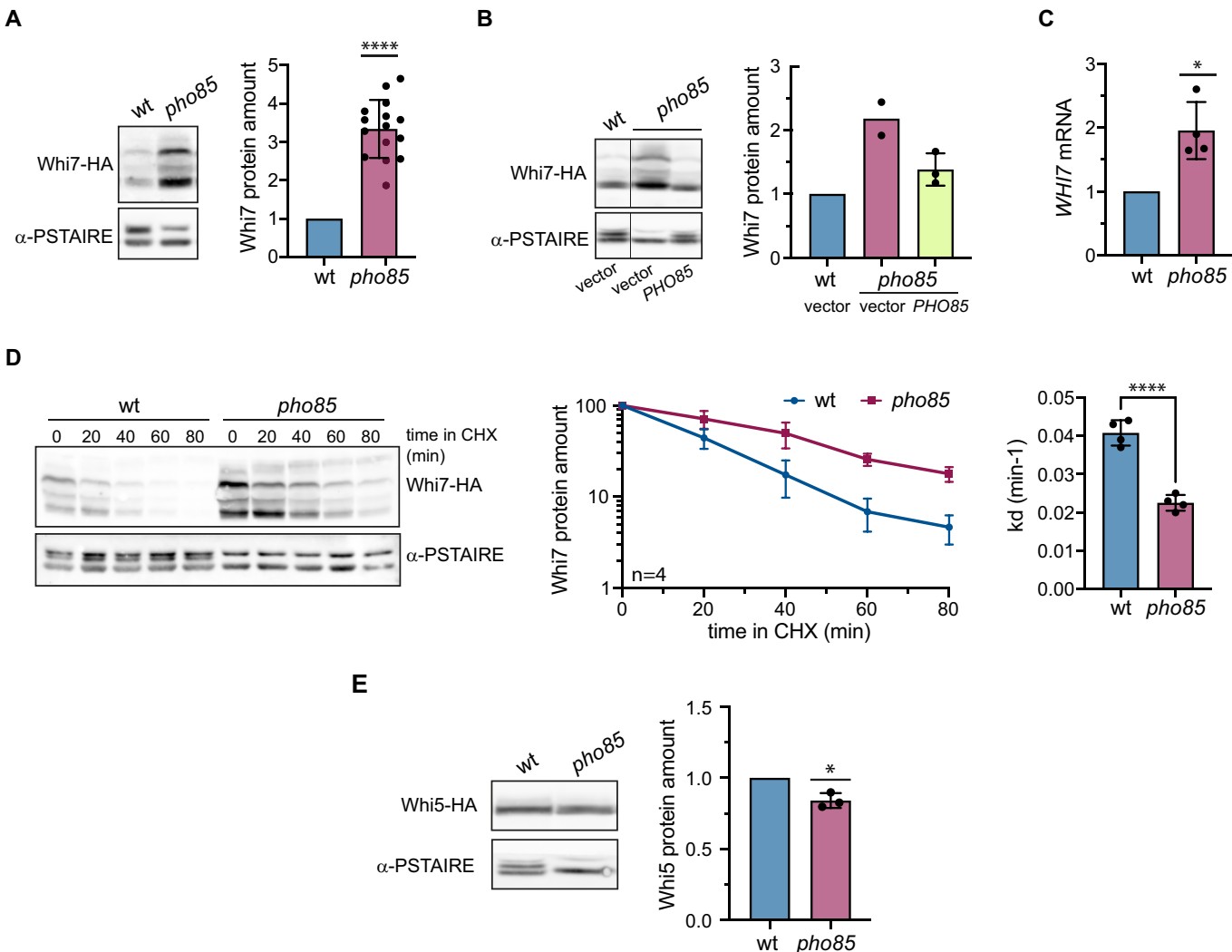

**Figure 1. The CDK Pho85 downregulates Whi7 at two levels: transcriptional repression and protein unstabilization.**

(A) Analysis of Whi7 protein levels in wild-type (JCY1728) and *pho85* (JCY2221) cells by western blot (*n* = 15). (B) Whi7 protein level was analyzed as in panel (A), in wild-type (JCY1728) and *pho85* (JCY2221) cells transformed with an empty vector or a centromeric plasmid containing the *PHO85* gene (*n* = 2–3). (C) The level of *WHI7* mRNA relative to *ACT1* mRNA was analyzed by quantitative RT-PCR in wild-type (W303-1a) and *pho85* (JCY2241) cells (*n* = 4). (D) Whi7 protein stability was analyzed by translational shut-off with cycloheximide (CHX). Wild-type (JCY1728) and *pho85* (JCY2221) cells were incubated in the presence of CHX 100 µg/mL and Whi7 protein level was analyzed at the indicated times after the addition of CHX by western blot. Graphs represent the quantification of the western blots (left panel) and the degradation rate constant ($k_d$) extracted from the slope of the linear regression obtained from the quantification of the individual western blots (right panel) of the indicated strains (*n* = 4). (E) Analysis of Whi5 protein level in wild-type (JCY2036) and *pho85* (JCY2282) cells (*n* = 3). Data information: (A–E) In the western blots, Cdc28 (lower band recognized by the α-PSTAIRE) is shown as loading control. In the graphs, data from the independent biological replicates are represented as mean ± s.d. In the bar graphs of panels (A–C) and (E), wild-type value is referred to as 1. ****$P \le 0.0001$ and *$P \le 0.05$; in panels (A),(C),(E), one-sample t-test was used to determine whether the mean was different to 1 (wt); in panel (D), two-tailed unpaired *t*-test was used. *n* = number of biological replicates. Source data are available online for this figure.

redundant role in the unstabilization of Whi7, since neither the inactivation of the Pcls cyclins (Fig. 2C) nor that of Pho80 (Fig. 2D) had a significant effect on their own, but the combination of the four mutations caused a stabilization of Whi7 protein similar to that observed in cells lacking *PHO85* (Fig. 2E).

One of the best-characterized targets of the Pho85-Pho80 signaling pathway is the Pho4 transcription factor, that becomes nuclear and drives the expression of phosphate-responsive genes upon Pho85-Pho80 inactivation (Carroll et al, 2001). However, we did not detect Pho4 binding to *WHI7* promoter (Fig. EV2A) or

induction of *WHI7* transcription under phosphate starvation (Fig. EV2B). In addition, Whi7 protein stability also remains roughly unaffected during phosphate starvation (Fig. EV2C), as occurs in normal conditions in the *pho80* mutant (Fig. 2D). Thus, Pho85-Pho80 do not regulate Whi7 levels in phosphate limitation.

In summary, we conclude that CDK Pho85 regulates Whi7 protein levels through its association with multiple cyclins; specifically, Pho85 regulates the expression of the *WHI7* gene through its association with Pho80 cyclin, while it controls the stability of the Whi7 protein through its association with Pho80 and Pcl1,2,9 cyclins.

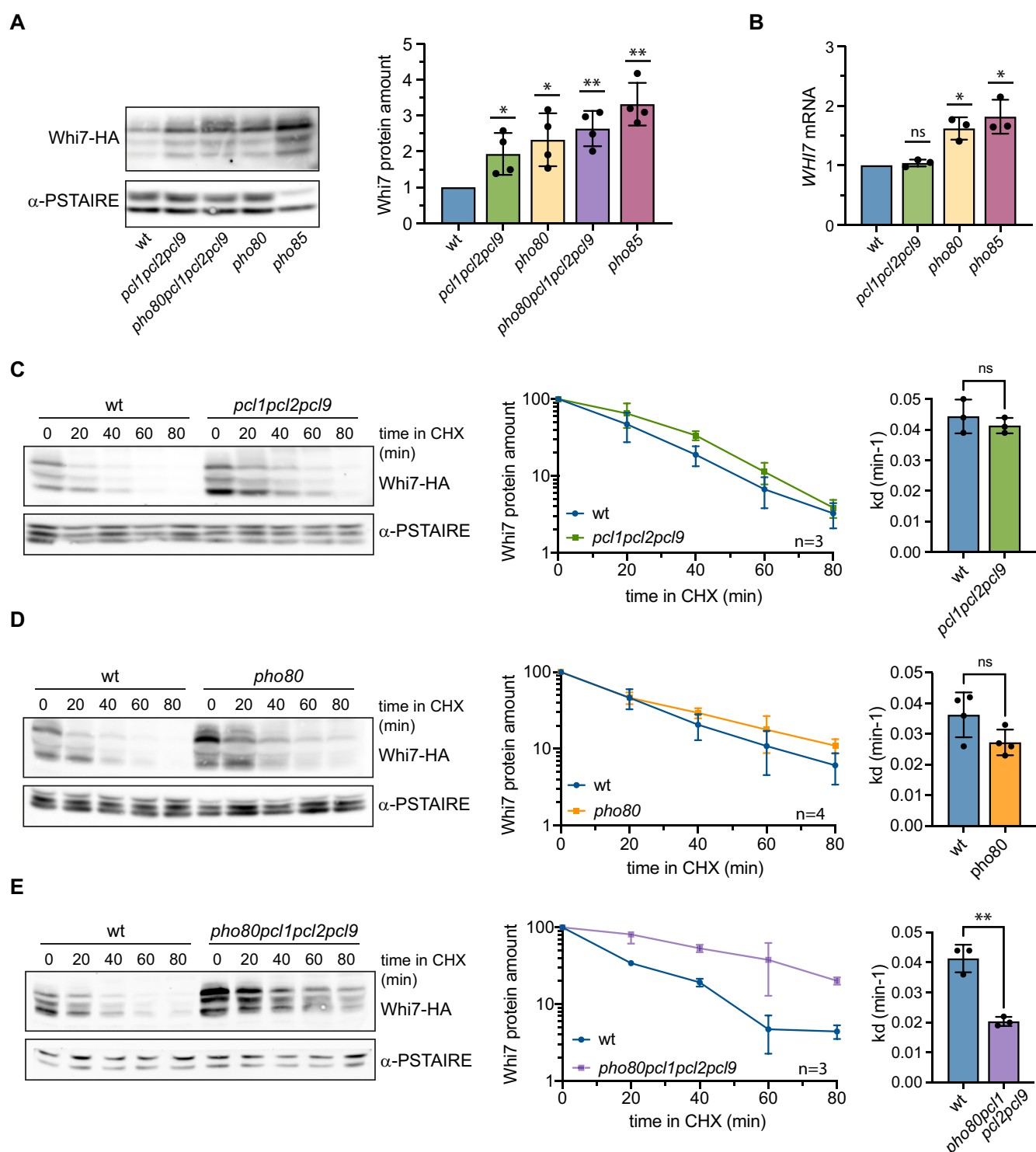

**Figure 2. The G1 Pcl1,2,9 and Pho80 cyclins mediate the effect of Pho85 on Whi7.**

(A) Analysis of Whi7 protein level in wild-type (JCY1728), *pho85* (JCY2221) and Pho85-associated cyclins mutant strains *pcl1pcl2pcl9* (JCY2256), *pho80pcl1pcl2pcl9* (JCY2337) and *pho80* (JCY2335) by western blot (n = 4). Wild-type value is referred to as 1. (B) The level of *WHI7* mRNA relative to *ACT1* was analyzed by quantitative RT-PCR in wild-type (W303-1a), *pcl1pcl2pcl9* (JCY2315), *pho80* (JCY2333) and *pho85* (JCY2241) cells (n = 3). Wild-type value is referred to as 1. (C–E) Whi7 protein stability was analyzed as described in Fig. 1D in the same strains used in panel (A). Graphs represent the quantification of the western blots (left panel) and the degradation rate constant (right panel) of the indicated strains. Data information: (A–E) In the western blots, Cdc28 (α-PSTAIRE) is shown as loading control. In the graphs, data from the independent biological replicates are represented as mean ± s.d. **$P \leq 0.01$; *$P \leq 0.05$; and ns, $P > 0.05$, in panels (A),(B), one-sample t-test was used to determine whether the mean was different to 1 (wt); in panels (C–E), two-tailed unpaired *t*-test was used. n = number of biological replicates. Source data are available online for this figure.

## The CDK Pho85 controls Whi7 protein stability through phosphorylation of Ser27 and Thr100

To test whether Whi7 is a direct target of Pho85, we performed co-immunoprecipitation assays using the kinase-dead version of Pho85 (Pho85$^{E53A}$) (Wanke et al, 2005). As shown in Fig. 3A, we were not able to detect interaction with the wild-type version of Pho85, as expected for quick and transient interactions between kinases and their targets. However, Whi7 physically interacted with Pho85$^{E53A}$, suggesting that the CDK Pho85 directly acts on Whi7 in vivo, as was also reflected in the above-commented phospho-proteomics analyses. To further confirm whether Whi7 is a direct target of Pho85, we performed an in vitro kinase assay. We took advantage of the fact that Whi7 phosphorylation can be easily detected by conventional electrophoresis and western blotting, due to the appearance of slow-migrating bands (Gomar-Alba et al, 2017). Whi7-HA was purified from yeast cells and treated with lambda phosphatase. The resulting non-phosphorylated Whi7 protein was used as a substrate for Pho85-TAP kinase purified from wild-type cells. Whereas no band shift was detected in the control sample, incubation with Pho85 resulted in phosphorylation of Whi7, as deduced from the appearance of slow-migrating bands (Fig. 3B). From all these results, we conclude that Whi7 is a bona fide substrate of Pho85 kinase.

Among the 12 consensus S/TP CDK sites present in Whi7, two of them, Ser27 and Thr100, adjust to the optimal phosphorylation consensus sequence for Pho85 (S/T-P-X-I/L). Because of that we carried out the same in vitro kinase assay with a version of Whi7 with the two Pho85 candidate sites, Ser27 and Thr100, mutated to Ala (hereafter called Whi7$^{S27A,T100A}$ or Whi7-ST2A). Importantly, the amount of phosphorylated isoforms in Whi7$^{S27A,T100A}$ was significantly reduced compared to the wild-type protein (Fig. 3B). This indicates that Pho85 phosphorylates Whi7 at Ser27 and Thr100 amino acids. Note that there are still slow-migrating bands in Whi7$^{S27A, T100A}$; this is consistent with the phosphoproteomics data and points to Pho85 affecting the phosphorylation of additional CDK consensus sites.

Next, we investigated whether Pho85 acts directly through the phosphorylation of these two residues to increase Whi7 protein instability. We analyzed the stability of Whi7 protein in wild-type cells carrying the phospho-null version of Whi7 (Whi7$^{S27A,T100A}$). As shown in Fig. 3C, abolishing the phosphorylation of Ser27 and Thr100 resulted in the stabilization of the Whi7 protein, with a Whi7 $k_d$ significantly reduced compared to that of wild-type protein and very similar to that observed upon Pho85 inactivation. No stabilization was detected in the case of the single mutants, suggesting that Pho85 is able to signal through both residues (Fig. EV3A). Interestingly, the phospho-mimic version of Whi7, in which both amino acids were replaced with Glu (Whi7$^{S27E,T100E}$ or Whi7-ST2E), turned highly unstable in pho85 mutant cells, thus bypassing the stabilization of Whi7 caused by the absence of Pho85 (Fig. 3C). Consistently with these observations, total Whi7 protein levels increase in Whi7$^{S27A,T100A}$ but not in pho85 Whi7$^{S27E,T100E}$, when compared with the wild-type (WHI7) (Fig. 3D). Finally, there is no additive effect of pho85 inactivation on Whi7$^{S27A,T100A}$ or Whi7$^{S27E,T100E}$ (Fig. EV3B–C), supporting that, at least in the control of protein stability, Pho85 is acting mainly through these two residues. Note that Whi7 $k_d$ in minimal media (Fig. 3, Figs. EV2 and EV3) is approximately half of that observed in rich media

(YPD, Figs. 1 and 2), suggesting that Whi7 stability increases under suboptimal nutrient conditions.

We previously reported that the F-box protein Grr1 drives the degradation of Whi7 via SCF$^{Grr1}$ ubiquitin ligase. To investigate whether the Pho85-dependent degradation of Whi7 has the same mechanistic bases, we performed co-immunoprecipitation assays with overexpressed Grr1ΔF-FLAG, a version of Grr1 able to bind substrates without promoting its degradation. As shown in Fig. 3E, the physical interaction between Whi7 and Grr1ΔF is lost in the case of Whi7$^{S27A,T100A}$. In conclusion, altogether these results support that Whi7 is a substrate for Pho85 kinase and that Pho85 mediates the degradation of Whi7 through the phosphorylation of Ser27 and Thr100, which is important for its recognition by the SCF$^{Grr1}$ ubiquitin ligase.

## Whi7 overexpression causes a G1 arrest in the absence of Pho85

Whi7 is functionally redundant with Whi5 in the control of G1/S (Gomar-Alba et al, 2017). Overexpression of the Whi5 repressor has been shown to be toxic in the pho85 mutant strain (Huang et al, 2009). Because of this, to further investigate the connection between the Start repressor Whi7 and Pho85, we assayed the effect of overexpressing WHI7 from the GAL1 promoter with β-estradiol in pho85 mutant cells. Although, as expected, there was no significant effect on cell growth in wild-type cells (Gomar-Alba et al, 2017), in the absence of Pho85 high levels of Whi7 severely impaired cell growth (Fig. 4A,B). Microscopy analysis of cells revealed a first cycle arrest at Start after induction of WHI7 expression with the accumulation of approximately 90% of single nucleated large unbudded cells (Fig. 4C). It can be expected that Whi7 causes this G1 arrest by repressing the Start transcriptional program. Indeed, we found that CLN2 gene expression was drastically repressed in pho85 cells overexpressing Whi7 (Fig. 4D). Furthermore, ectopic expression of CLN2 from the S. pombe adh1 promoter restored the proliferation of pho85 cells overexpressing Whi7 (Fig. 4E). We conclude that, in the absence of PHO85, high levels of Whi7 cause repression of the Start transcriptional program, leading to G1 arrest. This is consistent with the former results showing that Pho85 negatively regulates Whi7 cellular levels, and reveals that Pho85 may be inhibiting Whi7 function as a Start repressor.

We next investigated the role of the Pho85 cyclins in this G1 arrest. We performed the same GAL1pr:WHI7 overexpression assays in pho80, pcl1, pcl2, and pcl9 single mutants and in pcl1pcl2pcl9 triple mutant. We found that the induction of Whi7 only caused G1 arrest in the absence of Pcl1, with no differences in the phenotypes observed when comparing pcl1 and pcl1,2,9 (Fig. 4F,G and Appendix Fig. S2).

## Pho85 promotes Start through the specific inhibition of Whi7

In wild-type cells, Whi5 has higher protein levels than Whi7, which could explain why Whi5 plays a more important role than Whi7 under normal conditions (Gomar-Alba et al, 2017). Unlike Whi5, Whi7 is induced upon cell wall stress and becomes the major Start repressor in this specific condition (Méndez et al, 2020). Since depletion of Pho85 increases Whi7, but not Whi5, protein levels, we asked whether this could lead to a more relevant role of Whi7 in the control of Start in pho85 mutant cells. To directly compare Whi5

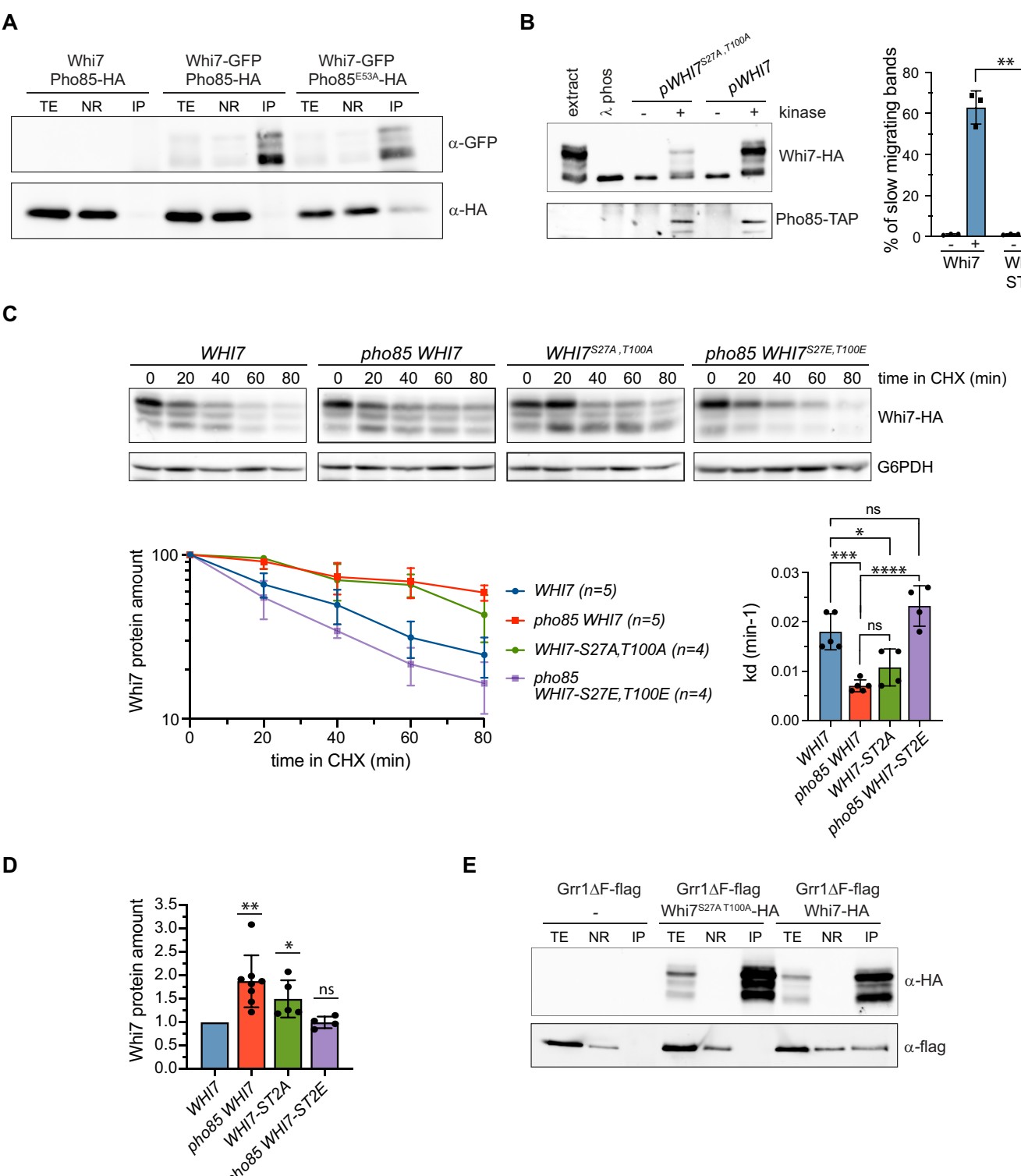

and Whi7 strength in repressing Start, we monitored their protein levels using the β-estradiol-*GAL* system, which in this case allowed us to express Whi7-GFP and Whi5-GFP in a β-estradiol dose-dependent manner and thereby determine the conditions under which the cells contain the same protein level of either Whi7 or Whi5 repressor. Moreover, the proteins can be expressed at a

moderate level, avoiding artifacts due to strong overexpression of the *GAL1* promoter in galactose. The expression of *GAL1pr:WHI7-GFP* or *GAL1pr:WHI5-GFP* at different doses of β-estradiol in both wild-type (Fig. EV4A) and *pho85* cells (Fig. EV4B) was measured by single-cell fluorescence microscopy in order to determine the optimal concentrations of β-estradiol to obtain equal amounts of

**Figure 3. Pho85 controls Whi7 protein stability through the phosphorylation of Ser27 and Thr100 amino acids.**

(A) Wild-type (W303-1a) or Whi7-GFP (JCY1746) cells were transformed with a centromeric plasmid expressing Pho85-HA or Pho85$^{E53A}$-HA. Whi7-GFP was immunoprecipitated from crude extracts and the presence of Whi7 and Pho85 in the total extract (TE), non-retained (NR) and immunoprecipitated (IP) fractions was determined by western blot. (B) Kinase in vitro assay of Whi7. Whi7-HA or the phospho-null version Whi7-HA$^{S27A,T100A}$ expressed from a centromeric plasmid was purified from cell lysates (*extract*) of *whi7* cells (JCY1819) and was then treated with λ protein phosphatase (λ *phos*) to obtain dephosphorylated Whi7 as substrate for the assays. Dephosphorylated Whi7 was incubated with purified Pho85-TAP (JCY2652) (+) or kinase buffer buffer (−). Whi7 phosphorylation, detected by the slow-migration electrophoretic bands, was analyzed by western blot and the % of slow migrating bands was quantified from three independent biological replicates. (C) Whi7 protein stability was analyzed as described in Fig. 1D in *whi7* cells (JCY1819) transformed with a centromeric plasmid expressing Whi7 (*WHI7*) or the S27A T100A phospho-null version of Whi7 (*WHI7*$^{S27A, T100A}$) and *whi7pho85* cells (JCY2475) transformed with Whi7 or the S27E T100E phospho-mimic version of Whi7 (*WHI7*$^{S27E,T100E}$). G6PDH is shown as loading control. Graphs represent the quantification of the western blots (*left panel*) and the degradation rate constant (right panel) of the indicated strains. (D) Analysis of Whi7 protein levels in t0 of the strains used in panel (C) ($n = 4$–8). (E) The *grr1* mutant strain (JCY1539) was co-transformed with a plasmid expressing a flag-tagged Grr1 lacking its F-box (Grr1ΔF) and either the Whi7-HA or Whi7-HA$^{S27A,T100A}$ plasmid or a control vector. Cells were grown on raffinose medium and transferred to galactose medium for 4 h. Whi7 was immunoprecipitated from crude extracts and the presence of Grr1ΔF and Whi7 in the total extract (TE), non-retained (NR) and immunoprecipitated (IP) fractions was determined by western analysis. Data information: (B–D) In the graphs, data from the independent biological replicates are represented as mean ± s.d. ****$P ≤ 0.0001$; ***$P ≤ 0.001$; **$P ≤ 0.01$; *$P ≤ 0.05$; and ns, $P > 0.05$, two-tailed unpaired *t*-test. $n$ = number of biological replicates. Source data are available online for this figure.

both proteins. Note that, unlike Whi5, Whi7 is a highly unstable protein, and it was necessary to use a higher dose of β-estradiol to compensate for protein degradation. In the case of wild-type cells, *WHI7* was induced using 7.5 nM β-estradiol whereas *WHI5* was induced using 1 nM β-estradiol to obtain cells with the same amount of protein (Fig. 5A,B). This expression had no significant effect on cell proliferation (Fig. 5C). However, the induction of Whi5 in wild-type cells caused an accumulation of cells in the G1 phase (Fig. 5D). This effect was not observed when Whi7 was expressed at the same level, consistent with the idea that Whi5 plays a more important role than Whi7 in Start repression under normal conditions (Gomar-Alba et al, 2017; Méndez et al, 2020). Note that the slight accumulation in G1 cells observed in cells overexpressing Whi5 (Fig. 5D) does not affect the doubling times (Fig. 5C), suggesting that cell cycle compensatory mechanisms must be acting, as occurs for other Start mutants that show changes in G1 duration without affecting their overall division time (Kumar et al, 2018).

In the case of *pho85* cells, induction of *WHI7* and *WHI5* with 10 nM and 1.25 nM β-estradiol respectively, resulted in cells expressing the same amount of Whi7 or Whi5 as determined by fluorescence microscopy and western blotting (Fig. 5E,F). Importantly, while Whi5 induction in the absence of Pho85 had no effect on cell proliferation, Whi7 induction caused a first cycle arrest in cell cycle progression in G1, with the accumulation of approximately 80% of cells as unbudded (Fig. 5G,H). Because these G1 cells contain the same total amount of either Whi7 or Whi5, and even higher amounts of Whi5 in the nucleus (Fig. 5I), this result clearly demonstrates that Whi7 plays a more important role than Whi5 as Start repressor when Pho85 is absent.

Previous studies have described that *pho85* mutant cells exhibit an elongated G1-phase (Wysocki et al, 2006). To characterize the contribution of Whi5 and Whi7 repressors to the G1 phase duration in the *pho85* mutant, the percentage of unbudded cells in asynchronous cultures was first determined. As it was previously reported (Gomar-Alba et al, 2017), the deletion of both *WHI5* and *WHI7* genes caused a decrease in the percentage of unbudded cells in the wild-type strain, reflecting a shortened G1-phase, which is consistent with their role as Start repressors, being the effect of Whi5 inactivation stronger than that of Whi7 inactivation. The inactivation of both repressors also resulted in a shortening of the G1 phase in *pho85* cells; however, in this case, the effect was stronger when inactivating Whi7 than when inactivating Whi5 (Fig. 6A). Next,

we extended this study to synchronized cells. As expected, after release from an α-factor induced G1-arrest, *pho85* cells manifested a delay in the activation of Start compared to wild-type cells (Fig. 6B). Interestingly, Whi5 inactivation did not rescue this delay, whereas Whi7 inactivation dramatically advanced the G1/S transition in *pho85* cells (Fig. 6B). It should be noted that the *whi7* mutation does not fully alleviate the *pho85* Start activation delay, suggesting that in addition to Whi7, Pho85 may control the G1/S transition through additional targets. In agreement with the role of Whi7 as a repressor of the Start transcriptional program, the expression of the *CLN2* gene was advanced in the *whi7pho85* double mutant when compared to *pho85* single mutant (Fig. 6C). These results clearly indicate that the Start repression in *pho85* cells is mediated mainly by Whi7 and not by Whi5.

## Pho85 phosphorylation of Whi7 directly restrains Whi7 association to promoters

We next analyzed whether Pho85 regulates the association of Whi7 and Whi5 to the Start gene promoters by Chromatin Immunoprecipitation (ChIP) experiments. We used a yeast strain carrying a tagged version of the two repressors, Whi7-Myc and Whi5-GFP, and assayed the association of Whi7 and Whi5 with the promoters of the *CLN2* cyclin and the cell wall genes *FKS1* and *MNN1* in both wild-type and *pho85* cells. For all three promoters, Whi7 binding increased approximately 5-fold upon Pho85 inactivation (Fig. 7A), whereas Whi5 binding was unaffected. We conclude that CDK Pho85 promotes Start, at least in part, by decreasing the association of Whi7 to the Start promoters.

The increased binding of Whi7 to promoters in the absence of Pho85 could be a direct consequence of the increased protein levels of the repressor. To test this hypothesis, we used the β-estradiol-*GAL* system to equally express Whi7 in the wild-type and *pho85* mutant cells. Strikingly, similar levels of Whi7-GFP protein resulted in an approximately 3–5-fold increase in the binding of Whi7 to *CLN2*, *FKS1*, and *MNN1* promoters in the *pho85* mutant compared to the wild-type strain (Fig. 7B). This clearly points out that Pho85 can directly restrain the potential of Whi7 as a repressor by inhibiting Whi7 binding to promoters, independently of the control of Whi7 protein levels.

As described above, Pho85 affects the stability of Whi7 through the phosphorylation of residues Ser27 and Thr100. To further ascertain whether Pho85 regulates Start through the

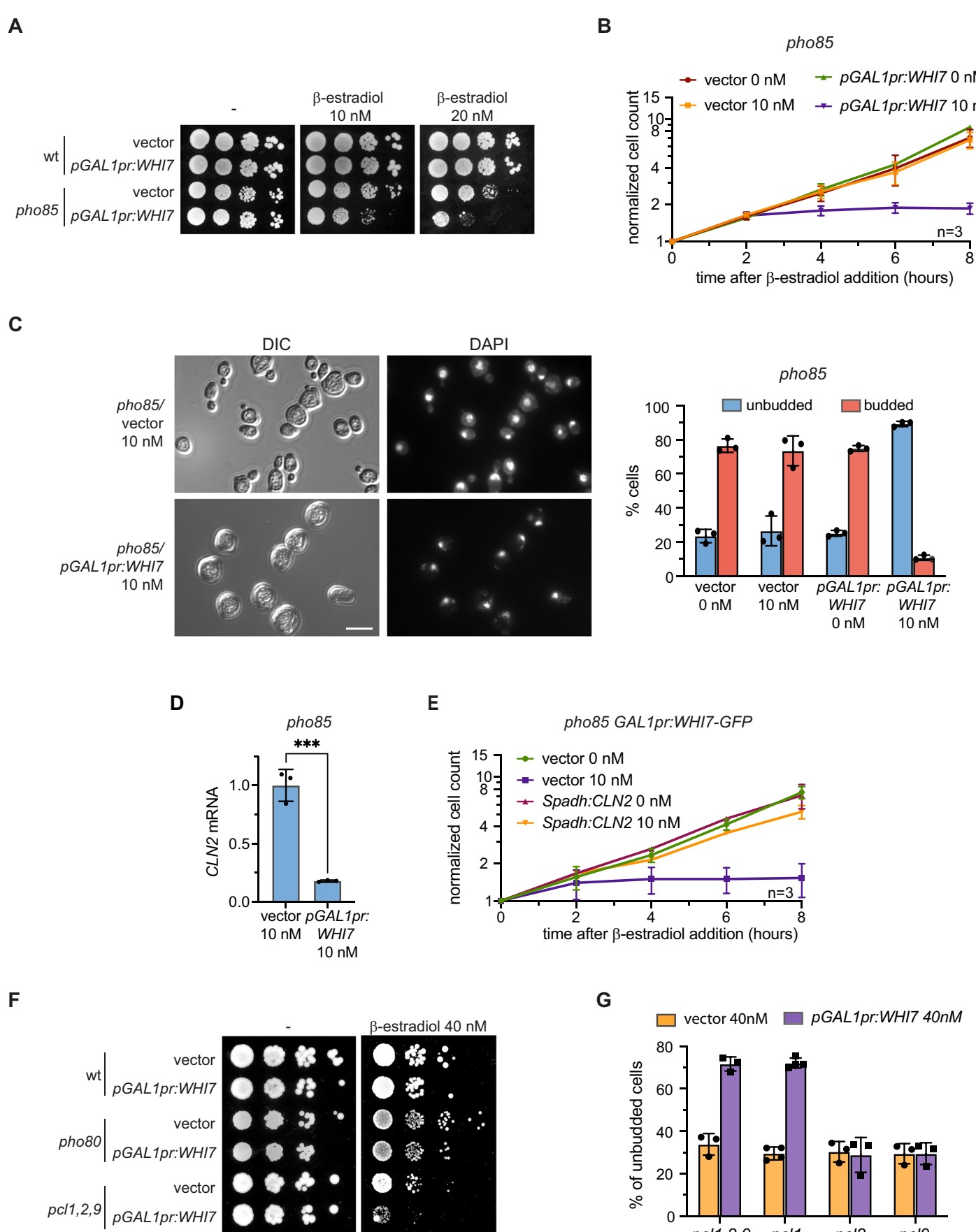

**Figure 4. *WHI7* overexpression causes a G1 arrest in the absence of Pho85.**

(A) 10-fold serial dilutions from wild-type (JCY2443) or *pho85* (JCY2486) cells carrying the β-estradiol hybrid transactivator (ADGEV) and transformed with an empty vector or a plasmid expressing *WHI7* under the control of the *GAL1* promoter (pGAL1pr:WHI7) were spotted onto SC plates with the indicated concentrations of β-estradiol and incubated at 25 °C for 3 days. The experiment was done three times with similar results. (B) Exponentially growing cells of *pho85* ADGEV (JCY2486) transformed with an empty vector or the pGAL1pr:WHI7-HA plasmid were incubated in the presence of 0 nM or 10 nM β-estradiol (to induce overexpression of *WHI7*) and the increase in cell number over time was analyzed. Cell number was normalized to 1. (C) Cell cycle distribution of *pho85* cells used in panel (B) after 4 h in 10 nM β-estradiol. Microscopy images of DIC and DAPI staining of DNA of *pho85* cells treated with 10 nM β-estradiol. Scale bar, 8 μm. Graph represents the percentage of unbudded and budded cells (*n* = 3). At least 200 cells were scored for each replicate, strain and condition. (D) The level of *CLN2* mRNA relative to *ACT1* mRNA was analyzed by quantitative RT-PCR in the same strains of panel B after 4 h incubation with 10 nM β-estradiol (*n* = 3). The mean of *pho85 vector* is referred to as 1. (E) Exponentially growing cells of *pho85 GAL1pr:WHI7-GFP ADGEV* (JCY2561) strain transformed with an empty vector or a plasmid constitutively expressing *CLN2* (*Spadh:CLN2*) were grown with 10 nM or without β-estradiol and the increase in cell number was analyzed. Cell number was normalized to 1. (F) 10-fold serial dilutions from wild-type (JCY2443), *pho80* (JCY2486) or *pcl1,2,9* (JCY2626) cells carrying the β-estradiol hybrid transactivator (ADGEV) and transformed with an empty vector or pGAL1pr:WHI7 were assayed as in panel (A). (G) Percentage of unbudded cells of *pcl1,2,9* (JCY2626), *pcl1* (JCY2831), *pcl2* (JCY2832), or *pcl9* (JCY2833) strains transformed with an empty vector or pGAL1pr:WHI7 after 4 h incubation with 40 nM β-estradiol (*n* = 3–4). At least 200 cells were scored for each replicate, strain and condition. Data information: In panels (B),(C),(D),(E) and (G), data from the independent biological replicates are represented as mean ± s.d. ***$P \leq 0.001$, two-tailed unpaired *t*-test. *n* = number of biological replicates. Source data are available online for this figure.

phosphorylation of Whi7, the effect of this phosphorylation on Whi7 association with promoters was investigated. We performed ChIP experiments with cells expressing wild-type, phospho-null (S27A, T100A) or phospho-mimic (S27E, T100E) version of Whi7. As shown in Fig. 7C, Whi7[S27A,T100A] presented increased binding to *CLN2* promoter, whereas in *pho85* cells the substitution of Ser27 and Thr100 to Glu drastically reduced the ability of Whi7 to associate with the promoters (Fig. 7D). Note that in both cases the experiment was carried out by adjusting the amount of immuno-precipitated Whi7 in each sample (Fig. 7C,D) to avoid any possible contribution caused by the difference in protein levels. We wondered whether mutation of Ser27 and Thr100 could affect the subcellular localization of Whi7. However, we observed the same cell cycle distribution in the phoshomutants than in the wild-type protein (Fig. EV5) and both, the cell cycle nuclear localization pattern and the nuclear levels of Whi7 in G1 cells remained unaffected (Fig. 7E,F), similarly of what we observed for *pho85* (Fig. EV1). Thus, the differences in promoter association found in *pho85*, Whi7 [S27A,T100A] and Whi7 [S27E,T100E] are not due to differences in Whi7 nuclear accumulation. Therefore, we finally investigated by co-immunoprecipitation assays whether the association of Whi7 with SBF was impaired when Whi7 was phosphorylated by Pho85. Indeed, this is the case, since the in vivo physical interaction between Swi4 and Whi7 was severely reduced in Whi7[S27E,T100E] (Fig. 7G). Taken together, these observations allowed us to conclude that Pho85, through the phosphorylation of Ser27 and Thr100, inhibits the intrinsic capacity of Whi7 to bind to promoters through Swi4. In summary, the CDK Pho85 acts as a brake on Whi7 function to promote Start.

## Discussion

We previously described that the CDK Cdc28 regulates Whi7 protein levels and its subcellular localization. Specifically, when the 12 CDK consensus sites of Whi7 are mutated to Ala, Whi7 becomes stable, constitutively nuclear and accordingly presents higher association to Start promoters. In this work we have extended our studies on the connection between Whi7 and CDKs, including the CDK Pho85. Phosphoproteomic analysis have shown that Whi7 exists in vivo as an hyperphosphorylated protein. Our *ad hoc* analysis on Whi7 is consistent with and extends previous global studies by other authors that identified, in aggregate, up to 17

phosphorylation sites in Whi7 under different growth conditions (see Lanz et al, 2021). This is a similar case to other Start transcriptional repressors. Indeed, up to 25 or 36 phosphorylated sites have been experimentally confirmed in different studies in the case of Whi5 or mammalian Rb, respectively (see Hasan et al, 2014). This massive phosphorylation reveals that these proteins could act as molecular hubs on which multiple kinases impinge. Of particular interest, Whi7/Srl3 was originally identified as a multi-copy suppressor of the DNA damage checkpoint mutant *rad53* (Desany et al, 1998) and we have identified phosphorylation of two consensus sites for the Mec1 and Tel1 checkpoint kinases (S222 and T225). This strengthens the connection of Whi7 function with the response to DNA damage, in agreement with a global analysis of the proteome done in this condition (Lanz et al, 2021). As regards CDK kinases, it is noteworthy that we have identified phosphor-ylation in 11 out of the 12 CDK consensus sites, which reinforces the relevance of Whi7 regulation by CDKs. Furthermore, the phosphoproteomic results from *pho85* cells strongly suggest redundancy between Cdc28 and Pho85 kinases in the phosphor-ylation of Whi7 at multiple CDKs sites.

Unlike Cdc28, Pho85 is not essential for cell cycle progression, suggesting that it could play a minor role in cell cycle regulation. However, genetic interactions revealed that Pho85 is required for the Start transition when this process is compromised by mutation in the G1 Cln cyclins (Measday et al, 1994; Espinoza et al, 1994), indeed pointing to an important contribution of this CDK in the control of G1/S transition. Subsequent studies began to unravel this connection, demonstrating that Pho85 acts as a positive regulator of Start by acting through several mechanisms: stabilization of Cln3 cyclin (Menoyo et al, 2013), inactivation of the transcriptional repressor Whi5 (Huang et al, 2009) and degradation of CKI Sic1 (Nishizawa et al, 1998). However, although some Pho85 targets have been described, a complete picture of the underlying molecular mechanisms by which Pho85 controls the cell cycle remains to be elucidated. Here, we provide further insight into the connection between Pho85 and the Start control, identifying a new target: the Whi7 transcriptional repressor. The in vivo and in vitro assays carried out in this work showed a direct interaction between Whi7 and Pho85 and phosphorylation of Whi7 driven by Pho85. Strikingly, when we examine α-factor synchronized cells, the G1 delay caused by the depletion of *PHO85* is largely abolished (although not completely, as expected given the aforementioned mechanisms) by the mutation of *WHI7*, suggesting that Whi7

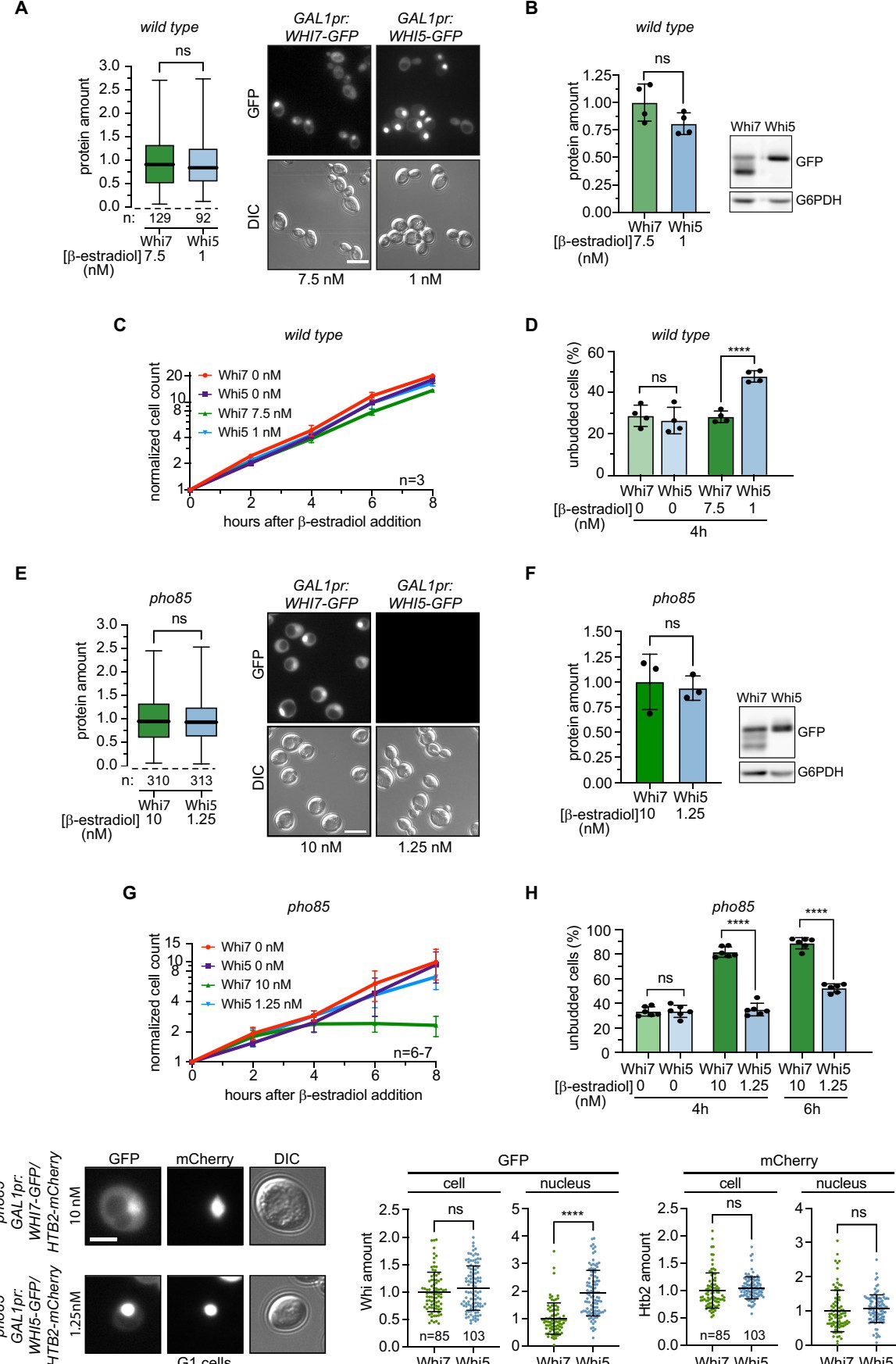

**Figure 5.  Whi7 is the main Start transcriptional repressor in the absence of Pho85.**

(A) Quantification of total fluorescence in single cells expressing Whi7-GFP (JCY2446) or Whi5-GFP (JCY2448) using the β-estradiol-*GAL* system. Different concentrations of β-estradiol are used to obtain similar levels of Whi7 and Whi5. To normalize fluorescence intensity, value 1 was given to the mean of Whi7-GFP. GFP fluorescence and bright field (DIC) microscopy images of Whi7-GFP and Whi5-GFP induced with 7.5 nM and 1 nM β-estradiol respectively are shown. Scale bar 8 μm. (B) Protein levels of Whi7-GFP and Whi5-GFP were determined by western blot in the same cells used in panel (A). G6PDH is used as loading control. A representative western blot image is shown. The mean of Whi7 is referred to as 1. (*n* = 4). (C) Exponentially growing cells of *GAL1pr:WHI7-GFP ADGEV* (JCY2446) and *GAL1pr:WHI5-GFP ADGEV* (JCY2448) were incubated in the presence of 7.5 nM (Whi7) or 1 nM (Whi5) β-estradiol and the increase in cell number over time was analyzed. Cell number was normalized to 1. (D) Cell cycle distribution of cells used in panel C after 4 h from the addition of β-estradiol (*n* = 4). At least 200 cells were scored for each replicate, strain and condition. (E–H) The same experiments in (A–D) were carried out in *pho85* mutant cells expressing Whi7-GFP (JCY2561) or Whi5-GFP (JCY2559). In this case, 10 nM or 1.25 nM β-estradiol was used to induce expression of the *WHI7* or *WHI5* gene respectively. In panel (E), scale bar 8 μm. In panel (F), *n* = 3. In panel (H), *n* = 6 and at least 200 cells were scored for each replicate, strain and condition. (I) Cells of the indicated strains (JCY2951 and JCY2978) were imaged by fluorescence microscopy and the levels of Whi7-GFP and Whi5-GFP were determined in G1 cells after 4 h of induction with the β-estradiol conditions used in panel (E). The histone Htb2 tagged with mCherry was used as control for nuclear levels. Total fluorescence intensity of GFP and mCherry signals was measured by segmentation of either the whole-cell in the DIC channel or the nuclear area in the mCherry channel. To normalize fluorescence intensity of GFP and mCherry, value 1 was given to the mean of Whi7-GFP strain. Scale bar 4 μm. Data information: In panels (A) and (E) boxes include 50% of data points, the line represents the median, and whiskers extend to maximum and minimum values. One of two independent experiments with similar results is shown. In panels (B),(C),(D),(F),(G),(H) data from the independent biological replicates are represented as mean ± s.d. In panel (I), mean ± s.d. of the single cells is shown. ****$P \le 0.0001$, and ns, $P > 0.05$, two-tailed unpaired *t*-test. ****. In panels (A),(E) and (I), *n* = number of cells. In panels (B),(C),(D),(F),(G),(H), *n* = number of biological replicates. Source data are available online for this figure.

inactivation is one of the main contributions by which Pho85 promotes cell cycle entry.

Pho85 negatively regulates Whi7 protein levels via two independent mechanisms. On one hand, Pho85 triggers Whi7 instability through its phosphorylation at Ser27 and Thr100, which promotes the interaction with SCF[Grr1] ubiquitin ligase. This mechanism involves many cyclins, as only the combined depletion of Pho80, Pcl1, Pcl2, and Pcl9 increases Whi7 stability similarly to *pho85* mutation. On the other hand, Pho85 downregulates the *WHI7* gene at transcriptional level. The fact that the *pho80* mutation increased *WHI7* mRNA levels to a similar extent as the *pho85* mutation suggests that Pho85 acts in combination with Pho80 cyclin to repress *WHI7* expression. Pho85-Pho80 inactivation triggers Pho4-dependent expression of *PHO* genes under phosphate limitation. However, we did not detect any association of the Pho4 transcription factor to the *WHI7* promoter in the absence of Pho85 or increased expression of *WHI7* upon phosphate starvation, suggesting that other transcription factors regulated by Pho85 must be involved. Consistent with this, a transcriptomic assay carried out with an analog-sensitive version of Pho85 showed upregulation of multiple stress response genes in a Pho4-independent manner when Pho85 was inactivated (Carroll et al, 2001). We hypothesize that the fact that Pho85 downregulates Whi7 levels by two independent mechanisms, protein instability and transcriptional repression, could reflect the need for (i) distinct mechanisms to robustly inactivate the Whi7 repressor and (ii) modulable and dynamic Whi7 regulation (e.g., via distinct cyclins) depending on cellular inputs.

The negative regulation of Whi7 by Pho85 restrains Whi7 ability to bind to the Start genes promoters, as confirmed by the increased binding in the *pho85* mutant compared to wild-type cells. Importantly, this regulation is also exerted through an additional mechanism unrelated to the control of Whi7 protein levels, because we found that in the absence of Pho85, Whi7 binding to promoters was increased even when Whi7 protein was adjusted to the same level in wild-type and *pho85* strains. Furthermore, immunoprecipitation of similar amounts of Whi7 and Whi7[S27E,T100E] in a *pho85* background resulted in decreased association with promoters in the case of the phospho-mimic version. These results reveal an additional layer of Whi7 regulation by Pho85, further than the control of protein levels, directly affecting Whi7 functionality as a

Start repressor. We discarded that Pho85 could affect Whi7 subcellular localization, because the subcellular distribution of Whi7 and the percentage of nuclear Whi7 remain unaffected in *pho85* and in the Whi7[S27A,T100A] and Whi7[S27E,T100E] phosphomutants. Rather, our results strongly support that Pho85 directly affects the intrinsic ability of Whi7 to associate with promoters. Indeed, we found that the phosphorylation of Ser27 and Thr100 impairs the association of Whi7 with the SBF subunit Swi4. In addition, it cannot be ruled out that additional mechanisms could affect the repressing activity of Whi7 once it is bound to the promoters. Future studies in this direction will help to unveil a more complete picture of how Pho85 modulates/inhibits Whi7-mediated transcriptional repression in cell cycle control.

Whi7 is a functional homologue of Whi5 (Gomar-Alba et al, 2017). Besides, Pho85 has been previously related to Whi5. Specifically, the phosphorylation of Whi5 by Pho85 helps to disrupt its interaction with histone deacetylases Hos3 and Rpd3 in order to activate the Start transcriptional program (Huang et al, 2009). Remarkably, Pho85 did not affect the protein levels of Whi5 or the Whi5 ability to associate with its target promoters. Thus, the new Pho85-mediated mechanisms reported herein are specific to the Whi7 transcriptional repressor. Indeed, two observations support the idea that the regulation of Whi7 must be physiologically dominant to explain how Pho85 regulates Start: (i) when both Start repressors are mildly overexpressed at the same level in *pho85* mutant, although Whi5 presented higher nuclear levels in G1 cells, Whi7, but not Whi5, caused a G1 arrest; (ii) the G1 delay of *pho85* cells after release from an α-factor arrest is dependent on Whi7 but not Whi5. In summary, all these observations indicate that, in the absence of Pho85, Whi7 has a more powerful role as a Start repressor than Whi5, and imply that cell cycle progression of wild-type cells relies in part on Whi7 inhibition exerted by Pho85.

We believe that the control of Pho85 on Whi7 may be particularly relevant to ensure cell cycle arrest and re-entry under stress conditions. In normal conditions, Cdk Pho85 is active and blocks cellular responses to unfavorable environmental conditions. Under non-optimal conditions, Pho85 signaling is inhibited, triggering the expression of stress genes (Carroll and O'Shea, 2002). This connection between Pho85 and the stress response was first characterized in the case of phosphate limitation. In optimal conditions, Pho85-Pho80 kinase inhibits the Pho4 transcription

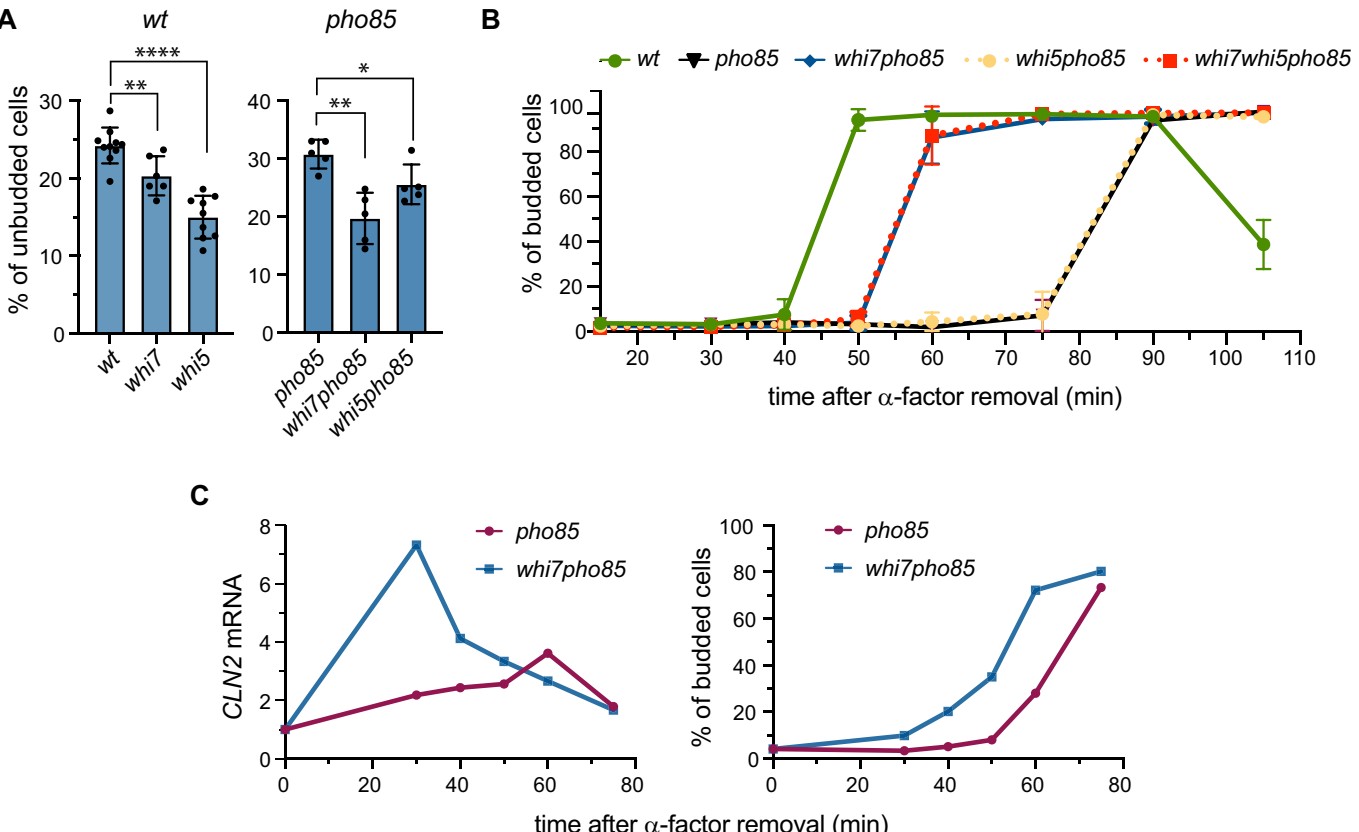

**Figure 6. The G1 delay of the *pho85* mutant is mediated by Whi7 and not by Whi5.**

(A) Cell cycle distribution of exponentially growing cells of wild-type (W303-1a), *whi7* (JCY1819), *whi5* (JCY1874), *pho85* (JCY2241), *whi7pho85* (JCY2475) and *whi5pho85* (JCY2538) strains (*n* = 5–10). Values show the mean ± s.d. At least 200 cells were scored for each replicate and strain. (B) Exponentially growing cultures of wt (W303-1a), *pho85* (JCY2241), *whi7pho85* (JCY2475), *whi5pho85* (JCY2538) and *whi7whi5pho85* (JCY2540) were arrested in G1 by incubation with α-factor for 3 h (>95% of cells arrested). After release from the arrest, cell cycle progression was monitored by bud emergence (indicative of Start activation) and the % of budded cells was scored at the indicated times (*n* = 3–5). Values show the mean ± s.d. from at least three independent experiments. At least 200 cells were scored for each replicate and strain. (C) Cultures of *pho85* (JCY2241) and *whi7pho85* (JCY2475) were synchronized with α-factor for 3 h as in panel (B) and the level of *CLN2* mRNA relative to *ACT1* was analyzed by quantitative RT-PCR after release from the arrest (left panel). In the same cells, the cell cycle progression was monitored and scored by bud emergence at the indicated times as in panel (B) (right panel). One of two independent experiments with similar results is shown. Data information: ****$P ≤ 0.0001$; **$P ≤ 0.01$, and *$P ≤ 0.05$, two-tailed unpaired *t*-test. *n* = number of biological replicates. Source data are available online for this figure.

factor, but when there is a shortage of phosphate in the medium, the Pho85-Pho80 complex is inhibited so that the hypophosphorylated Pho4 can activate the expression of the genes responsible for phosphate uptake (Kaffman et al, 1994; O'Neill et al, 1996). In the following years, as the cellular response to other stresses was elucidated, this regulatory strategy emerged as a common pattern in the axis Pho85-stress response. Thus, the cellular response to calcium stress relies on the regulation of the transcriptional factor Crz1 (responsible for the expression of calcium signaling genes) by Pho85-Pho80 activity (Cyert, 2003; Sopko et al, 2006), or the response to amino acids deficiency involving the inhibition of the transcription factor Gcn4 (responsible for the expression of genes related with amino acid biosynthesis) by Pcl5-Pho80 (Bömeke et al, 2006). Pho85 also acts through non-transcriptional mechanisms to restrain the response to unfavorable conditions, such as the inhibition of glycogen synthase Gsy2 by Pho85-Pcl8 and Pho85-Pcl10 in conditions where hyperaccumulation of glycogen is not needed (Huang et al, 1998).

It is well known that cell cycle progression is halted under stress conditions. In the light of the idea that Pho85 activity is inhibited in order to release the cellular response to distinct stresses and due to the positive role of Pho85 in cell cycle progression, a scenery emerges in which Pho85 could be a key player for cell cycle arrest (and/or adaptation) under different stress conditions by integrating information on environmental conditions with the cell cycle regulation. In fact, this has been described for several stresses. For instance, phosphate limitation causes Cln3 unstabilization through inhibition of Pho85 (Menoyo et al, 2013) and in response to DNA damage, Pho85 promotes G1-checkpoint adaptation by reinitiating the cell cycle by repressing Pho4- and Swi5-dependent gene expression and the induction of Sic1 proteolysis (Wysocki et al, 2006). Pho85 also plays an important role in inducing cell cycle arrest in response to stress caused by nitrogen starvation through phosphorylation of the Ssa1 chaperone, which leads to Cln3 cyclin downregulation (Truman et al, 2012). The *WHI7* gene is induced in different stress conditions (Waern and Snyder, 2013; Gasch et al, 2000; García et al, 2004) and our group has

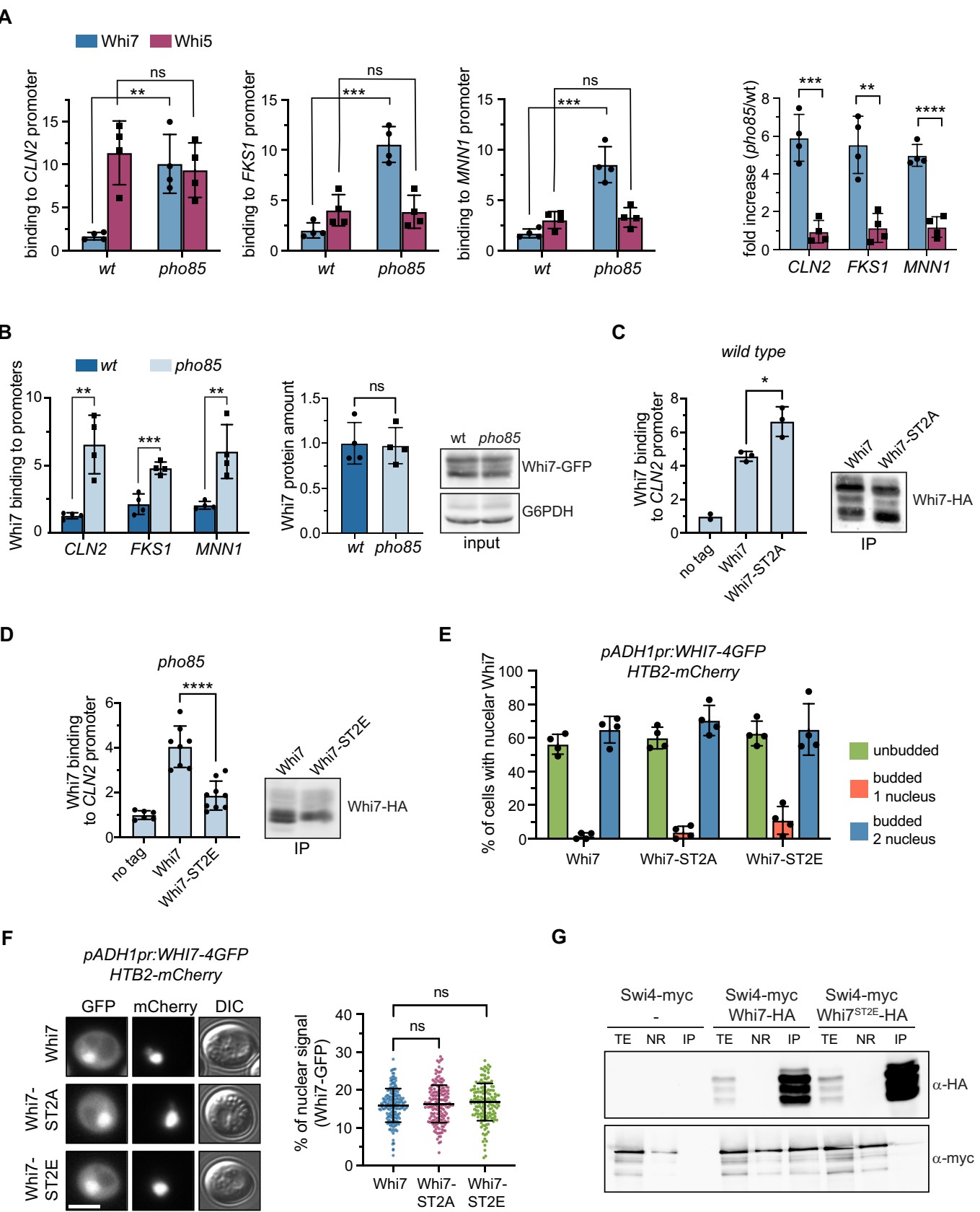

◀ **Figure 7.  Whi7 phosphorylation by Pho85 directly restrains Whi7 association to promoters, independently of Whi7 protein levels.**

(A) Whi7 and Whi5 association to the *CLN2*, *FKS1*, and *MNN1* gene promoters was analyzed by chromatin immunoprecipitation (ChIP) assays in wild-type (JCY2553) or *pho85* (JCY2556) cells expressing both *WHI7-myc* and *WHI5-GFP* ($n = 4$). Graph in the right represents the Whi7 and Whi5 binding fold increase (*pho85*/wt) to the indicated gene promoters. (B) ChIP assays of Whi7 binding to the indicated gene promoters in wild-type (JCY2446) and *pho85* (JCY2561) cells expressing Whi7-GFP at similar levels using the β-estradiol-*GAL* system ($n = 4$). In this case, wild-type cells were treated with β-estradiol 0 nM and *pho85* cells with β-estradiol 2.4 nM (left panel). For each ChIP assay, the protein levels of Whi7-GFP in wild-type and *pho85* cells was determined by western blot to confirm similar levels of induction. In the western blot, *G6PDH* is shown as loading control. A representative western blot image is shown. The mean of the wild-type is referred to as 1 (right panel). (C) Whi7-HA association to *CLN2* promoter was analyzed by ChIP in wild-type cultures (W303-1a) transformed with an empty vector (no tag), or a centromeric plasmid that expresses Whi7 or Whi7$^{S27A, T100A}$ ($n = 2$–3) (left panel). The experiment was carried out in conditions that guaranteed a similar level of Whi7 and Whi7$^{S27A, T100A}$ proteins in the immunoprecipitated samples to avoid effects caused by difference in protein levels (see text). A representative western blot of immunoprecipitated Whi7-HA showing similar amounts of proteins is shown (right panel). (D) Whi7-HA and Whi7$^{S27E, T100E}$-HA association to *CLN2* promoter was analyzed by ChIP in cultures of *pho85* (JCY2241) as in panel (C) ($n = 6$–9). (E) The percentage of cells with nuclear Whi7 was scored in the different cell-cycle phases in *whi7* cultures carrying Htb2-mCherry (JCY2981) transformed with a plasmid expressing *WHI7-4GFP*, *WHI7 $^{S27A, T100A}$-4GFP* or *WHI7 $^{S27E, T100E}$-4GFP* under the control of the *ADH1* promoter ($n = 4$). Htb2-mCherry was used as nuclear marker. At least 200 cells were scored for each replicate and strain. (F) Cells of the same strains used in panel (E) were imaged by fluorescence microscopy and the levels of Whi7-GFP were determined in G1 cells. Total fluorescence intensity of Whi7-GFP was measured by segmentation of either the whole-cell in the DIC channel or the nuclear area in the mCherry channel and the percentage of nuclear signal was determined in G1 single cells. Scale bar 4 µm. (G) Cells expressing a myc-tagged Swi4 (JCY1976) were transformed with either the Whi7-HA or Whi7-HA$^{S27E, T100E}$ plasmid or a control vector. Whi7 was immunoprecipitated from crude extracts and the presence of Swi4 and Whi7 in the total extract (TE), non-retained (NR) and immunoprecipitated (IP) fractions was determined by western analysis. Data information: In panels (A–E) data from the independent biological replicates are represented as mean ± s.d. In panel (F) mean ± s.d. of the single cells is shown. In the ChIP assays, the mean of the no tag strain is referred to as 1. ****$P \le 0.0001$; ***$P \le 0.001$; **$P \le 0.01$; *$P \le 0.05$; and ns, $P > 0.05$, two-tailed unpaired *t*-test. In panels (A–E), $n =$ number of biological replicates. In panel (F), $n =$ number of cells. Source data are available online for this figure.

demonstrated that Whi7 is the main Start transcriptional repressor in cell wall-stressed cells (Méndez et al, 2020). Interestingly, Pho85 is also involved in the cell wall stress response (Huang et al, 2002). Given the negative regulation of Whi7 by Pho85 described here, it is tempting to speculate that Whi7 regulation by Pho85 could be a major mechanism for the regulation of the cell cycle not only in cell wall stress, but also in other stresses that impinge on Pho85 activity. Reinforcing this, *WHI7* is related to genotoxic stress, as it was identified as a suppressor of the DNA damage checkpoint *rad53* mutant (Desany et al, 1998), and, as mentioned, Pho85 is involved in the control of the cell cycle after DNA damage (Wysocki et al, 2006). Future work will be necessary to identify the specific stress conditions in which the Pho85-Whi7 connection is relevant to ensure G1 arrest and cell cycle re-entry.

The CDK Pho85 is an ortholog of mammalian Cdk5, with 56% identity at the amino acid level. In fact, Cdk5 is able to carry out many functions of Pho85 when overexpressed in yeast, associating with Pho85 cyclins to constitute an active kinase and phosphorylating bona fide Pho85 substrates. Reciprocally, Pho85 expressed in mammalian cells associates with Cdk5 activators to form a functional kinase complex (Huang et al, 1999; Nishizawa et al, 1999; Huang et al, 2007). All these observations indicate that Pho85 and Cdk5 functions are conserved during evolution, and suggest that Pho85 and Cdk5 share common regulatory features. Unlike other CDKs, it was thought that Cdk5 plays a minimal role in the regulation of cell cycle progression in proliferative cells under normal conditions. Instead, Cdk5 activity is mostly present in the nervous system, where it governs virtually all aspects of brain development and neuronal physiology (Allnutt et al, 2020; Pao and Tsai, 2021). During the last years it has become clear that aberrant cell cycle re-entry in post-mitotic neurons, the ultimate consequence of which is apoptotic cell death, is a common feature in the development of neurodegenerative disorders such as Alzheimer's disease, Parkinson's disease, amyotrophic lateral sclerosis, and Huntington's disease (Allnutt et al, 2020; Requejo-Aguilar, 2023; Gupta et al, 2021; Xia et al, 2019). Remarkably, Cdk5 plays a central role in this cell cycle reactivation. Under normal conditions, Cdk5 associates with its activators p35 and p39, playing physiological roles during neurogenesis and neuronal function. For example,

Cdk5 can directly suppress cell cycle progression in neurons through the inactivation of the G1/S transcription factor E2F1-DP1 (Zhang et al, 2010). However, under pathological conditions, a truncated version of p35 named p25, is produced. Association to p25 causes Cdk5 hyperactivation and mislocalization. This redirects Cdk5 activity to aberrant targets that stimulate cell cycle re-entry in post-mitotic neurons, leading to neuropathological events (Gupta et al, 2021; Allnutt et al, 2020; Requejo-Aguilar, 2023; Joseph et al, 2020). Thus, the hyperactivation of Cdk5 is associated with neurodegenerative diseases.

Relevant to this work, one of the known targets of Cdk5 is the G1/S repressor Rb. In neurodegenerative disorders, hyperphosphorylation of Rb protein causes the activation of the Start transcriptional program and the consequent cell cycle re-entry that contribute to neuronal death (Joseph et al, 2020). Importantly, Cdk5 phosphorylates Rb when neuronal death is triggered by the overexpression of p25 in a neuronal cell line (Hamdane et al, 2005). In addition, Cdk5 can phosphorylate Rb protein in mouse brain during development and in one-year-old mice adults, although only when Cdk5 is overactivated (as it occurs in pathological conditions) phosphorylates Rb to the level that allows cell cycle initiation (Futatsugi et al, 2012; Liu et al, 2017; Sharma and Sicinski, 2020). On the other hand, Cdk5 is also generally dysregulated in various cancer cells (Liu et al, 2017; Sharma and Sicinski, 2020; Shupp et al, 2017; Pozo and Bibb, 2016), and, importantly, the Rb protein is a downstream target of Cdk5 in neuroendocrine thyroid cancer (Pozo et al, 2013). All these observations enhance the relevance of the results presented here: the characterization of how Pho85 controls the G1/S repressor Whi7 could help to better understand the mechanism underlying cell cycle activation by Cdk5 in neurodegenerative disorders or in cancer.

## Methods

### Strains and plasmids

Yeast strains used in this work are shown in Appendix Table S2. Gene deletions, promoter replacements and insertions of

C-terminal tags were generated by standard one-step PCR-based methods (Longtine et al, 1998; Janke et al, 2004). The *ADH1pr:GAL4-ER-VP16* transactivator (the DNA binding domain of Gal4 fused to the hormone binding domain of the human estrogen receptor and the activation domain of the viral transcriptional activator VP16, *ADGEV*) was integrated in the genome (Louvion et al, 1993) for the induction of *GAL1pr:WHI5* or *GAL1pr:WHI7* with β-estradiol. Centromeric plasmids pWHI7-HA and pGAL1:-WHI7-HA are a gift from Dr. M. Aldea (Yahya et al, 2014). The pPHO85 plasmid expressing *PHO85* under the control of its own promoter is a gift from Dr. Erin K. O'Shea (Carroll et al, 2001). Plasmids *tetO7-HA²-PHO85* (pVW883) and *tetO7-HA²-PHO85^{E53A}* (pVW884) are a gift from Dr. Claudio de Virgilio (Wanke et al, 2005). Plasmids pWhi7^{S27A}, pWhi7^{T100A}, pWhi7^{S27A,T100A}, and pWhi7^{S27E,T100E} were obtained by site-directed mutagenesis (Promega) in pWHI7-HA. pYES-Grr1ΔF-FLAG allows to overexpress a Grr1 protein lacking its F-box, such that it was still able to bind substrates without promoting its degradation. The *Spadh1:CLN2* centromeric plasmid expresses the Cln2 cyclin under the control of the mild *adh1* promoter from *S. pombe*.

Cells were grown in exponential conditions (below $1 \times 10^7$ cells/mL) at 25 °C in standard yeast extract-peptone-dextrose (YPD) medium or synthetic complete (SC) medium with 2% glucose. For phosphate starvation experiments, low-phosphate SC media was prepared by using phosphate-depleted yeast nitrogen base supplemented with KCl (Formedium) with 2% glucose. Where indicated, cells were incubated in the presence of 15 μg/ml α-factor (GenScript) or 1–40 nM β-estradiol (Sigma).

## Western blot analysis

Approximately $1 \times 10^8$ cells were collected, resuspended in 100 μL water and, after adding 100 μL 0.2 M NaOH, they were incubated for 5 min at room temperature. Cells were collected by centrifugation, resuspended in 50 μL of Laemmli Sample Buffer, and incubated for 5 min at 95 °C. Extracts were clarified by centrifugation, and equivalent amounts of protein were resolved in an SDS-PAGE gel and transferred onto a nitrocellulose membrane (GE Healthcare). The primary antibodies used were: monoclonal anti-HA peroxidase 3F10 antibody (Roche Diagnostics, Cat. No: 12013819001) diluted 1:5000; monoclonal anti-c-myc 9E10 antibody (Roche Diagnostics, Cat. No: 1667149) diluted 1:5000; monoclonal anti-GFP (Roche Diagnostics, Cat. No: 11814460001) diluted 1:5000; monoclonal anti-Cdc2 p34 (PSTAIRE) sc-53 (Santa Cruz Biotechnology Inc.) diluted 1:2000; polyclonal anti-G6PDH (Thermo Scientific, Cat.No: BS-6989R) diluted 1:20,000 and polyclonal anti-TAP antibody (Invitrogen, Cat. No: CAB1001) diluted 1:5000. Blots were developed with anti-mouse IgG and anti-rabbit IgG Horseradish Peroxidase conjugate (Thermo Fisher Scientific, Cat. No: 170-6516 or 31460 respectively) diluted 1:20,000 using the Supersignal West Femto Maximum Sensitivity Substrate (Thermo Scientific). Bands were quantified with a ImageQuantTM LAS 4000 mini biomolecular imager (GE Healthcare). Routinary controls were used to ensure the linearity of the signal in all the range of the distinct bands, and when needed, images with different exposures were also quantified to analyze images with similar intensity in wild-type and mutant strains.

## Gene expression analysis (RT-PCR)

Approximately $2 \times 10^8$ cells were broken in a FastPrep Precellys24 (Bertin technologies). Total RNA was extracted using Qiagen RNeasy kit according to the manufacturer's protocol. RNA was quantified with NanoDrop 2000 Spectrophotometer (Thermo Scientific). 10 μg of RNA were incubated with DNase (Roche) and after DNase inactivation and incubation with oligo dT, cDNA was obtained with reverse transcriptase Maxima Reverse Transcriptase (Thermo Scientific). cDNA was analyzed in triplicate by quantitative RT-PCR with DNA Engine Peltier Thermal Cycler (BioRad) using the NZYSpeedy qPCR Green Master Mix. mRNA levels of genes of interest were quantified relative to *ACT1* mRNA by the ΔCt method.

## Protein stability assays

To evaluate Whi7 stability, 100 μg/mL cycloheximide was added to exponentially growing cells. Samples were harvested at the indicated times and protein decay was analyzed by western blot analysis. Bands were quantified with ImageQuant LAS 4000 mini biomolecular imager (GE Healthcare).

## Kinase in vitro assay

To purify Pho85-TAP, $4 \times 10^{10}$ exponential cells were collected, washed with cold water and resuspended with one volume of lysis buffer (0.1 M NaCl, 50 mM Tris-HCl pH 7.5, 1.5 mM $MgCl_2$, 0.1% Triton X-100, 1 mM DTT, 1 mM PMSF and Protease Inhibitors Cocktail Set III (Calbiochem)). Cells were lysed by vortexing with glass beads for 20 min at 4 °C. The extract was then clarified by centrifugation first at 4000 rpm for 10 min and second at 12,000 rpm for 90 min. The sample was supplemented with glycerol to 5% (v/v). The protein purification was performed by incubating the extract with SepharoseTM 6 Fast Flow IgG beads (GE Healthcare) for 1 h at 4 °C. The beads were collected by centrifugation at 1800 rpm for 3 min at 4 °C and transferred to a Mobicol Classic (MoBiTec) column. The beads were then washed with 10 mL of lysis buffer without protease inhibitors and the elution was made overnight at 16 °C by adding 2 mg/mL of TEV protease.

For the in vitro protein kinase assays, Whi7-HA was purified from cells lysed with lysis buffer (0,1% Triton X-100, 250 mM NaCl, 5 mM EDTA, 50 mM Tris-HCl pH 8, 1 mM PMSF and Phosphatase Inhibitor Cocktail Set II, EDTA-Free (Calbiochem)) and vortex for 15 min at 4 °C. The extract was treated with λ phosphatase (New England Biolabs) at 30 °C for 30 min for dephosphorylation. The extract was then incubated in rotation for 30 min at 4 °C with dynabeads previously bound to HA-probe (F-7) antibody (Santa Cruz Biotechnology Inc. Cat. No: SC-7392) for 30 min at 4 °C. Beads were then washed three times in PBS (150 mM NaCl, 40 mM $Na_2HPO_4$, 10 mM $NaH_2PO_4$) containing 0.01% (v/v) Tween 20 and twice with the 2X kinase reaction buffer (50 mM Tris-HCl pH 7.5, 1 mM DTT, 10 mM MgCl2, 1 mM EDTA, 50 μM ATP and Phosphatase Inhibitor Cocktail Set II EDTA-Free (Calbiochem)). Kinase assay was performed with 45 μL of 2X kinase buffer and 45 μL of the purified Pho85-TAP, that were added directly to the Whi7-bound beads and incubated at 30 °C for

60 min. Kinase assays were stopped by placing the tubes on ice. Samples were supplemented with Laemmli buffer, boiled for 5 min at 95 °C and phosphorylation was analyzed by SDS-PAGE and western blot.

## Fluorescence microscopy

Conventional fluorescence microscopy was carried out with an Axioskop 2 microscope (Zeiss). The images were captured with an AxioCam MRm camera (Zeiss) and AxioVision v4.7 software (Zeiss). Image analysis and fluorescence intensity quantifications were done using ImageJ software (https://imagej.net/software/fiji/). Imaged cells were segmented using DIC acquisitions with the aid of YeastMate (Bunk et al, 2022) and ImageJ. A custom ImageJ macro segmented the nuclear signal (Htb2-mCherry), and after manual correction of ROI, total fluorescence of the mCherry and GFP channels was automatically determined for all the individual cells. After background substraction, the amount of Whi7-GFP and/or Whi5-GFP (total fluorescence intensity) was quantified in the whole cell and in the nuclear area, and the percentage of nuclear Whi7 relative to total Whi7 was calculated in single cells.

## Chromatin immunoprecipitation analysis

Approximately $5 \times 10^8$ cells were cross-linked by a 15 min incubation after the addition of formaldehyde to 1% (v/v) to the growth medium, followed by a 5 min incubation after the addition of glycine to 125 mM. Cells were washed and resuspended in 300 µL of lysis buffer (50 mM HEPES-KOH pH 7.9, 40 mM NaCl, 1 mM EDTA, 1% (v/v) Triton X-100, 0.1% (w/v) sodium deoxicholate, 1 mM PMSF, 1 mM benzamidine and Protease Inhibitor Cocktail Set III (Calbiochem)) and lysed by vortexing with glass beads for 30 min at 4 °C. Lysis buffer was supplemented to a final volume of 600 µL, chromatin was then fragmented by sonication and the sample was centrifuged at 12,000 rpm for 15 min. 20 µL from the supernatant was collected as a control of whole-cell extract (input) and the remaining was incubated with orbital rotation for 2 h at 4 °C with Dynabeads Protein G (Invitrogen) previously bound to the antibody. Beads were then washed 4 times in PBS (150 mM NaCl, 40 mM Na$_2$HPO$_4$, 10 mM NaH$_2$PO$_4$) containing 0.02% (v/v) Tween 20. Elution of bound protein was carried out twice with 40 µL of 50 mM Tris-HCl pH 8.0, 10 mM EDTA, 1% (w/v) SDS, by heating at 65 °C for 8 min. Cross-linking was reverted by overnight incubation at 65 °C with shaking. The eluted sample was digested for 90 min at 37 °C with 0.33 mg/mL proteinase K and DNA was purified with the High Pure PCR product purification kit (Roche Diagnostics). Co-immunoprecipitated DNA was analyzed in triplicate by quantitative PCR in a DNA Engine Peltier Thermal Cycler (Bio Rad) using the NZYSpeedy qPCR Green Master Mix. DNA analyzed are fragments from *CLN2*, *FKS1*, and *MNN1* promoters and an intergenic region as a control. Whi7 or Whi5 binding values indicate the specific enrichment of the investigated promoter fragments in the immunoprecipitated sample compared to the whole-cell extract (input) using the intergenic region as a control, calculated with the ΔΔCT method. Values are relative to the no tag control strain (value of 1 equivalent to no specific enrichment).

## Phosphoproteomic assays

Wild-type and *pho85* mutant cells expressing *GAL1pr-WHI7-HA* were grown overnight to exponential phase in the presence of 5 nM of β-estradiol. Approximately $4 \times 10^{10}$ cells were collected. Cells were washed and resuspended in 300 µL of lysis buffer (50 mM HEPES-KOH pH 7.9, 40 mM NaCl, 1 mM EDTA, 1% (v/v) Triton X-100, 0.1% (w/v) sodium deoxicholate, 1 mM PMSF, 1 mM benzamidine and Protease Inhibitor Cocktail Set III protease inhibitors EDTA-Free (Calbiochem)) and lysed by vortexing with glass beads at 4 °C. The supernatant was collected by centrifugation at 12,000 rpm for 9 min and incubated with orbital rotation for 2 h at 4 °C with Dynabeads Protein G (Invitrogen) previously bound to the HA-probe F-7 antibody (Santa Cruz Biotechnology). Beads were then washed two times in PBS (150 mM NaCl, 40 mM Na$_2$HPO$_4$, 10 mM NaH$_2$PO$_4$) containing 0.02% (v/v) Tween 20. Elution of bound protein was carried out twice with 25 µL of 50 mM Tris-HCl pH 8.0, 10 mM EDTA, 1% (w/v) SDS, by heating at 65 °C for 8 min.

Protein was quantified by Machery Nagel (Invitrogen) according to the manufacturer's instructions. After cysteine alkylation with IAM, extracts (ca 10 µg) were cleaned with SP3 protocol to eliminate detergent previous to in solution protein digestion with Trypsin (Müller et al, 2020). The digested samples were submitted to IMOC phospho enrichment with TiO2 as described (Thingholm et al, 2006).

Tandem mass spectrometry analysis (LC-MS/MS) was performed in a Tims TOF flex mass spectrometer (Bruker). The eluted peptides in the Evosep One system were ionized in a captive Spray with 1700 V at 200 °C and analyzed in a ddaPASEF mode. The TIMS section was operated with a 100 ms ramp time and a scan range of 0.6–1.6 Vs cm$^{-2}$. One cycle was composed of 1 MS scan followed by 10 PASEF MS/MS scans. MS and MS/MS spectra were recorded from *m/z* 100 to 1700. Singly charged ions were excluded from the precursor ions based on their *m/z* and 1/K0 values. The quadrupole isolation width was set to 2 Da. The TIMS elution voltage was calibrated linearly to obtain the reciprocal of reduced ion mobility (1/K0) using three selected ions (*m/z* 622, 922, and 1222) of the ESI-L Tuning Mix (Agilent, Santa Clara, CA, USA).

Three biological replicates (one of them with two technical replicates) for each wild-type and *pho85* mutant strains were assayed. Analysis of data was implemented in the FragPIPE v2.0 pipeline for a suite of computational tools enabling comprehensive analysis of mass spectrometry-based proteomics data. The MSFragger protein identification algorithms were used to search for phosphopeptides with the workflow LFQ-phospho standard parameters. In addition, the pipeline was used for a combined MS1 quantitation by IonQuant and the MaxLFQ procedure (Cox et al, 2014). Peptides and phosphopeptides from Whi7/Srl3 identified in each sample are listed in Dataset EV1. Peptide were identified with a confidence of >95%. The phosphorylation sites in Whi7/Srl3 identified in each sample are listed in Dataset EV2. A stringent high-confidence localization threshold was applied, the identified phosphosites being considered only when the best localization probability was >0.85. Phosphosite identified in at least two of the three biological replicates were assigned as positive. For the comparison of the relative abundance of phosphorylated CDK sites, we calculated in each sample the relative weight of the

MaxLFQ intensity for each phosphorylated CDK site referred to the total intensity of the phosphorylated non-CDK sites in the sample.

## Miscellaneous

The number of cells in growth profile experiments was analyzed starting from early exponentially growing cells ($1 \times 10^6$ cells/mL) after brief sonication in the particle count Beckman Coulter. Protein co-immunoprecipitation assays were performed as previously described (Gomar-Alba et al, 2017).

## Statistical methods and reproducibility

Graphs and statistical analysis (two-tailed unpaired *t*-test and one-sample *t*-test) were performed with GraphPad Prism and R software. For all the strains used in this work, at least two independent clones were tested with similar results. All data were obtained from at least two independent biological replicates, and each experiment was repeated on at least two different days. For single cell experiments, 85–200 cells were counted for each replicate.

# Data availability

The mass spectrometry proteomics data have been deposited to the ProteomeXchange Consortium via the PRIDE partner repository with the dataset identifier PXD046448.

# Peer review information

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

## Acknowledgements

We are grateful to Dr. Manuel Mendoza and all members of the Cell Cycle Lab for helpful comments. The proteomic analysis was performed in the proteomics facility of SCSIE University of Valencia. This proteomics laboratory is a member of Proteored. This study was supported by the Spanish Government and co-financed by ERDF from the European Union, grant number PID2020-119793GB-I00 to JCI, and by the Generalitat Valenciana, grant number CIGE2021-093 to MGA. CRC is a recipient of a Predoctoral Fellowship from the Spanish Government (PRE2018-083178) and MSB is a recipient of a contract Investigo from the Generalitat Valenciana (INVEST/2022/203).

## Author contributions

**Cristina Ros-Carrero**: Conceptualization; Validation; Investigation; Methodology; Writing—original draft; Writing—review and editing. **Mihai Spiridon-Bodi**: Software; Formal analysis; Validation; Investigation; Methodology; Writing—review and editing. **J Carlos Igual**: Conceptualization; Supervision; Funding acquisition; Validation; Visualization; Writing—original draft; Project administration; Writing—review and editing. **Mercè Gomar-Alba**: Conceptualization; Formal analysis; Supervision; Funding acquisition; Validation; Visualization; Writing—original draft; Project administration; Writing—review and editing.

## Disclosure and competing interests statement

The authors declare no competing interests.

# Expanded View Figures

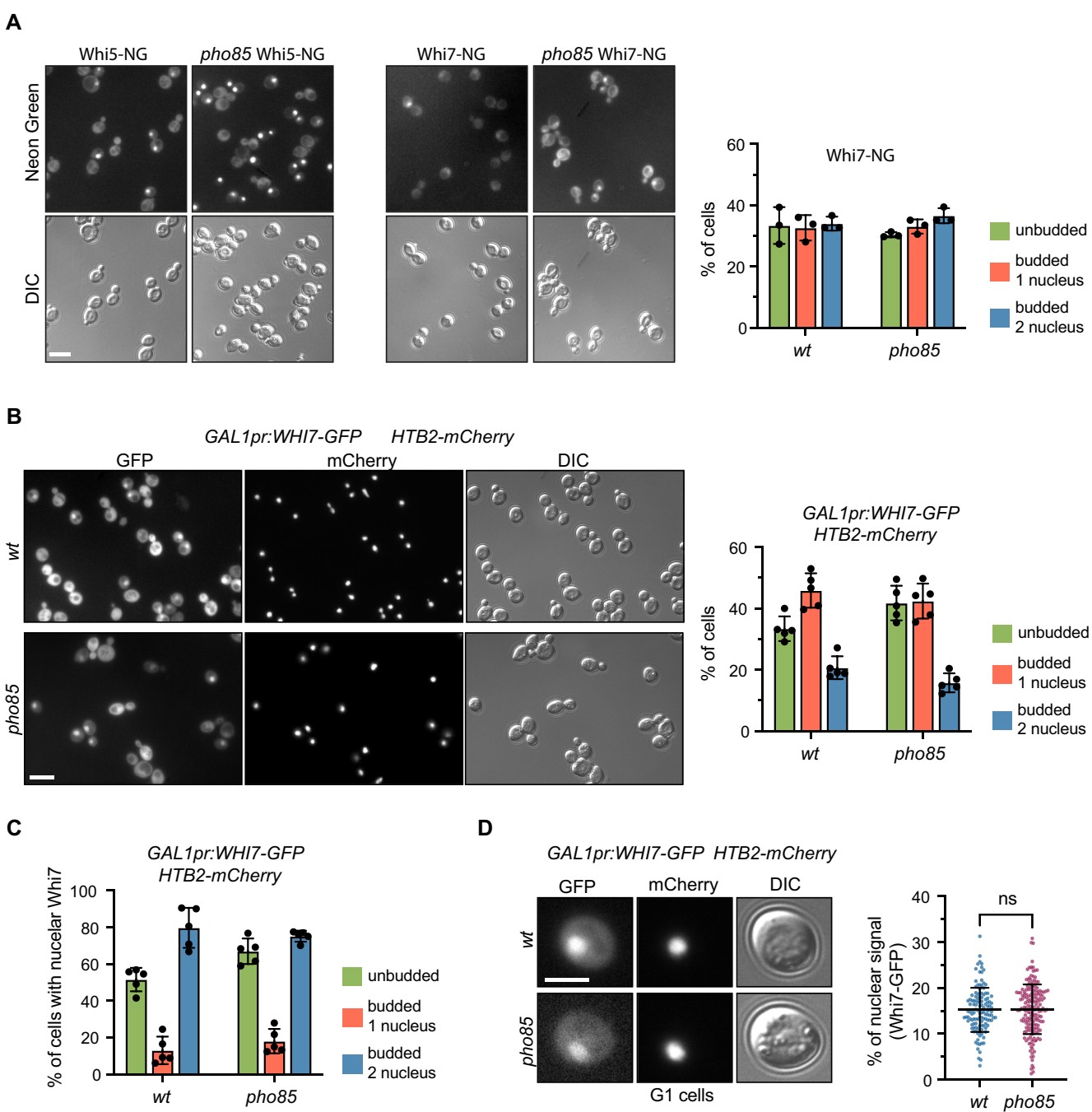

**Figure EV1. The subcellular localization of Whi7 is not regulated by the CDK Pho85.**

(A) Exponentially growing cells of the mNeonGreen-tagged Whi7 or Whi5 in wild-type (JCY2497 and JCY2495, respectively) and *pho85* (JCY2499 and JCY2501, respectively) strains were analyzed by conventional fluorescence microscopy. GFP and DIC channels are shown. Scale bar, 8 μm (*left panel*). Cell cycle distribution of wild-type and *pho85* cells carrying Whi7-NG (n = 3) (*right panel*). (B) Exponentially growing cells of the indicated strains (JCY2941 and JCY2951) were incubated with 1 nM β-estradiol for 1 h to obtain low Whi7-GFP induction. Cells were imaged by fluorescence microscopy. Scale bar, 8 μm (*left panel*) and the cell cycle distribution of wt and *pho85* cells carrying Whi7-GFP and Htb2-mCherry was sored (n = 5) (*right panel*). (C) In the same strains and conditions used in panel (B), the percentage of cells with nuclear Whi7 was scored in the different cell-cycle phases (n = 5). Htb2-mCherry was used as a nuclear marker. At least 200 cells were scored for each replicate and strain. (D) Cells of the same strains and conditions used in panel (B) were imaged by fluorescence microscopy and the levels of Whi7-GFP were determined in G1 cells. Total fluorescence intensity of Whi7-GFP was measured by segmentation of either the whole-cell in the DIC channel or the nuclear area in the mCherry channel and the percentage of nuclear signal was determined in G1 single cells. Scale bar 4 μm. Data information: In panels (A–C) data from the independent biological replicates are represented as mean ± s.d. In panel (D), mean ± s.d. of the single cells is shown. ns, $P > 0.05$, two-tailed unpaired *t*-test. In panels (A–C), n=number of biological replicates. In panel (D) n = number of cells.

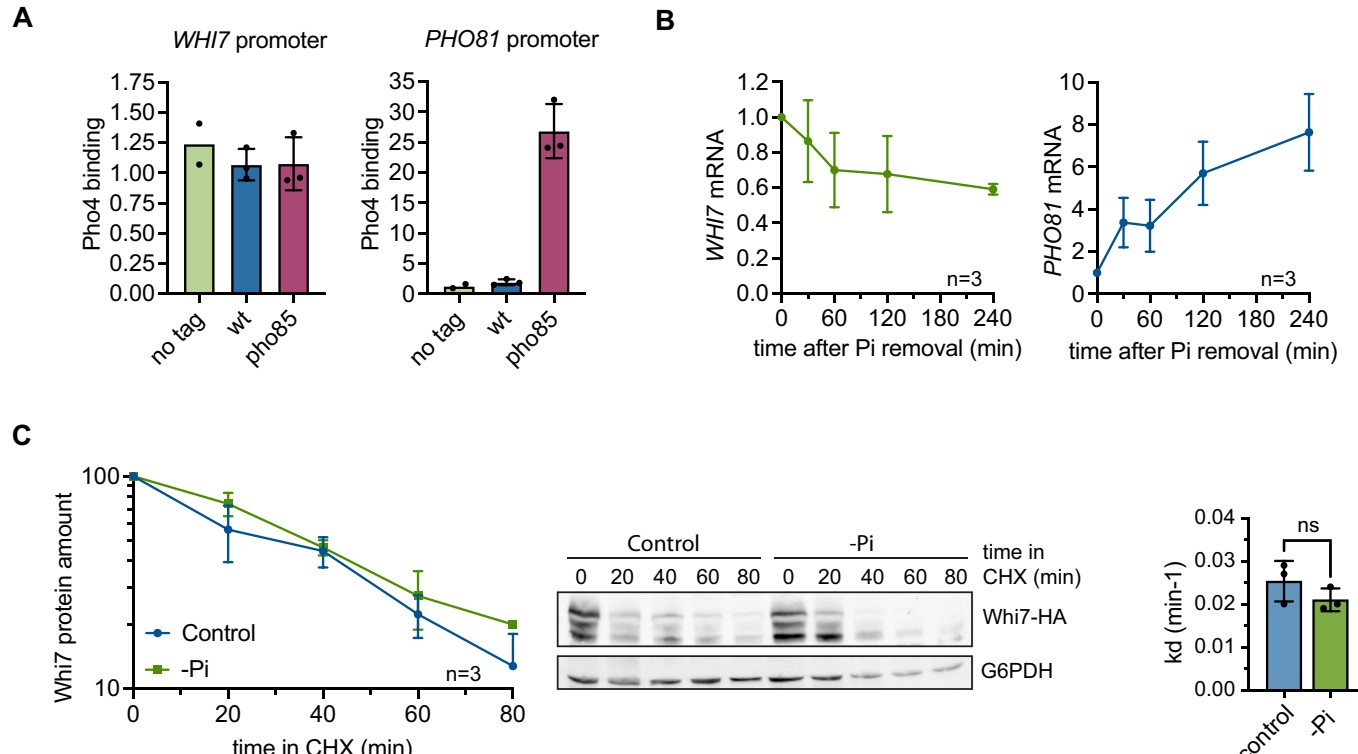

**Figure EV2. Whi7 is not regulated by phosphate starvation conditions.**

(A) Pho4 association to the *WHI7*, and *PHO81* gene promoters was analyzed by ChIP assays in wild-type (JCY2503) or *pho85* (JCY2505) cells expressing *PHO4-GFP* (n = 2–3). (B) The level of *WHI7* and *PHO81* mRNA relative to *ACT1* was analyzed by quantitative RT-PCR in wild-type (W303-1a) cells at the indicated times after phosphate (Pi) removal. Time 0 value is referred to as 1. (C) Whi7 protein stability was analyzed by translational shut-off with cycloheximide (CHX). Wild-type (JCY1728) cells were incubated for 3 h in either SC rich media (control) or SC low phosphate media (-Pi), CHX 100 µg/mL was added and Whi7 protein level was analyzed at the indicated times after the addition of CHX by western blot. Graphs represent the quantification of the western blots (left panel) and the degradation rate constant (right panel) of the indicated conditions. *G6PDH* is shown as loading control. Data information: In panels (A–C) data from the independent biological replicates are represented as mean ± s.d. ns, *P* > 0.05, two-tailed unpaired *t*-test. *n* = number of biological replicates.

    

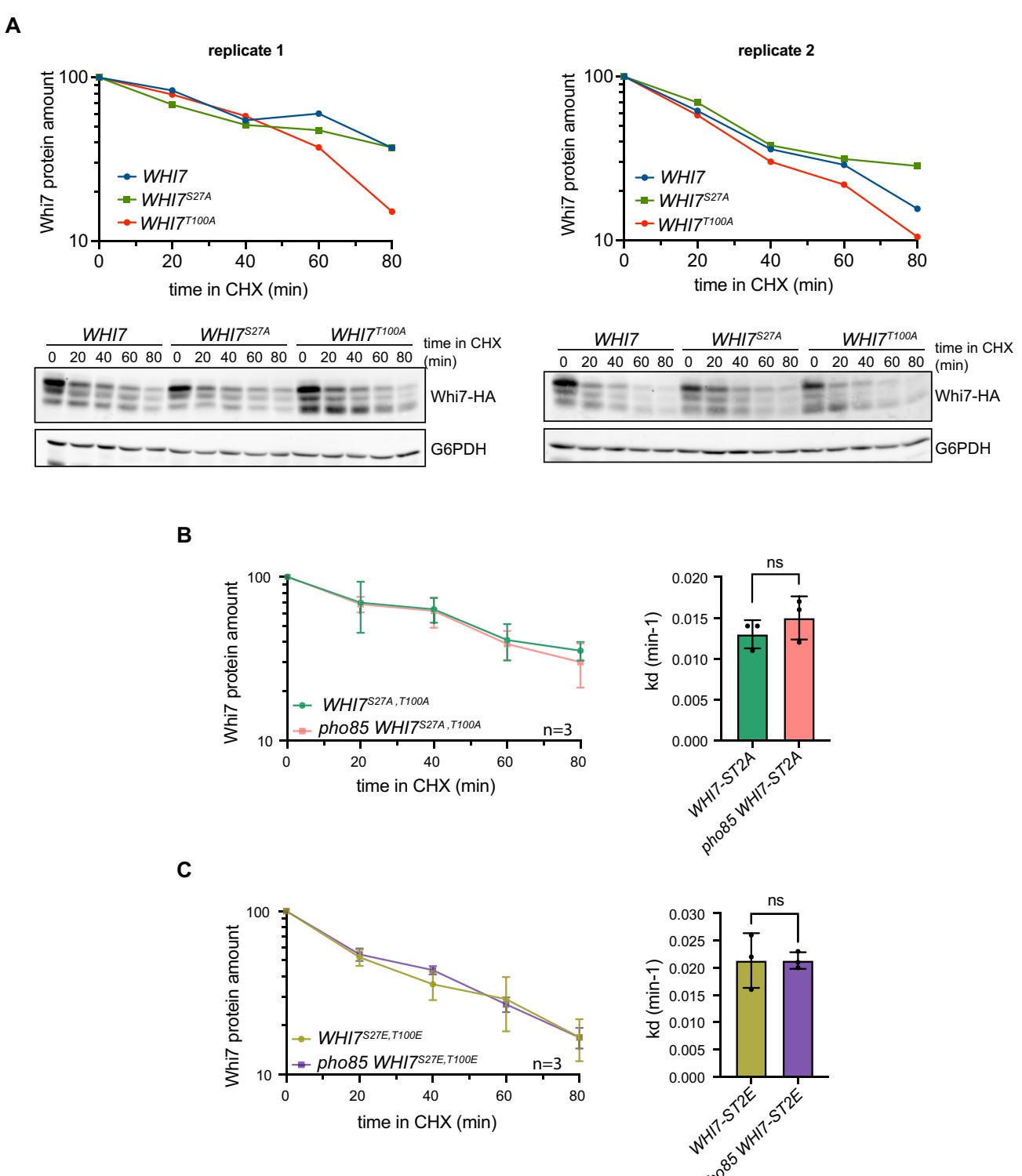

**Figure EV3. Analysis of the Ser27 and Thr100 mutations on Whi7 protein stability.**

(A) Whi7 protein stability was analyzed by translational shut-off with cycloheximide (CHX) in *whi7* cells (JCY1819) transformed with a centromeric plasmid expressing *WHI7*, *WHI7^S27A^* or *WHI7^T100A^*. Cells were incubated in the presence of CHX 100 μg/mL and Whi7 protein level was analyzed at the indicated times after the addition of CHX by western blot. *G6PDH* is shown as loading control. Graphs represent the quantification of two independent western blots (*n* = 2). (B,C) Whi7 protein stability was analyzed by translational shut-off as in panel (A) in wild-type and *pho85* cells carrying *WHI7^S27A, T100A^* or *WHI7^S27E, T100E^* expressed from a centromeric plasmid. Graphs represent the quantification of the western blots (left panel) and the degradation rate constant (right panel) of the indicated conditions. Data information: In panels (B,C) data from the independent biological replicates are represented as mean ± s.d. ns, *P* > 0.05, two-tailed unpaired *t*-test. *n* = number of biological replicates.

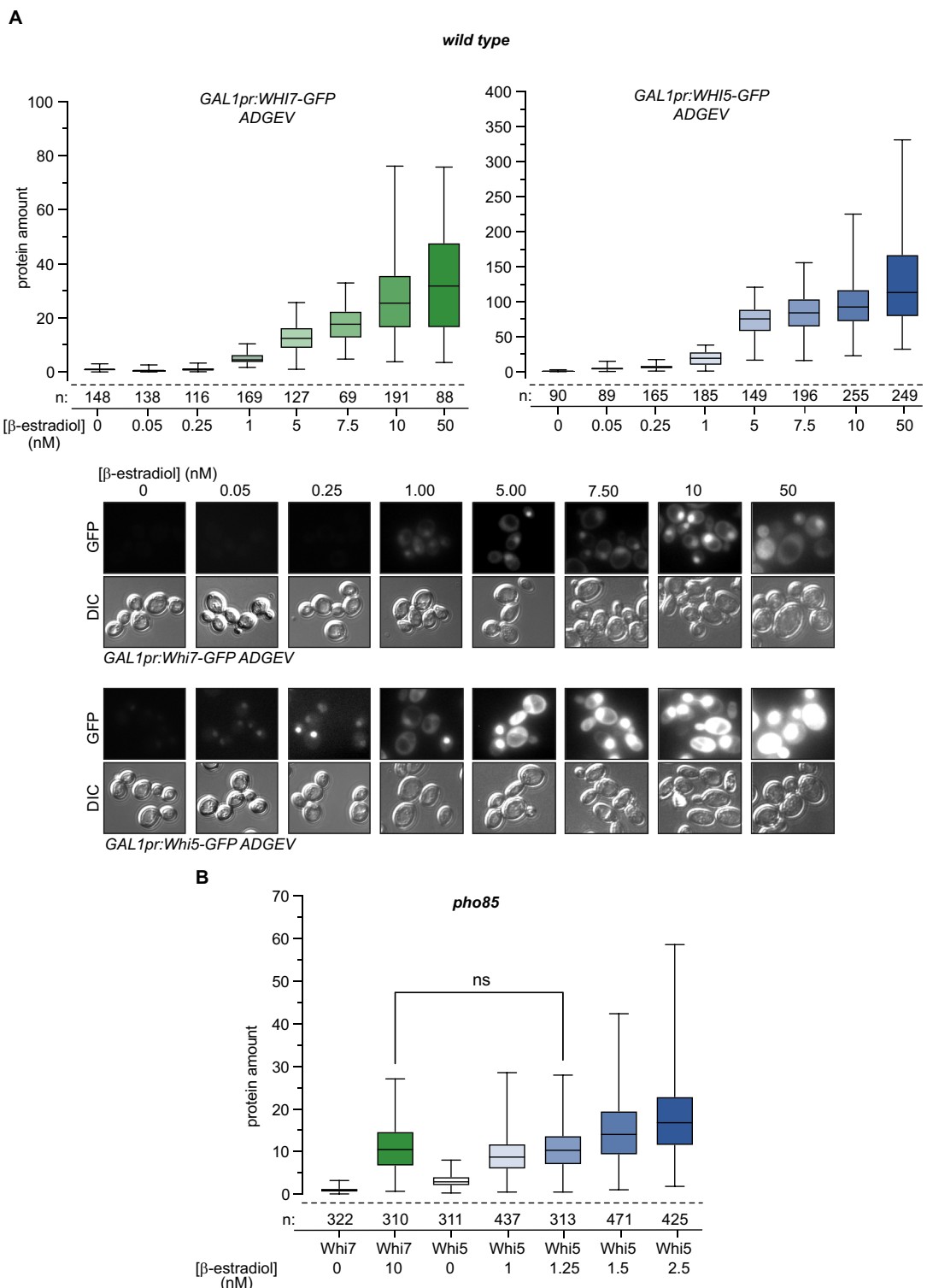

**Figure EV4. Analysis of Whi7 and Whi5 protein levels at different β-estradiol concentrations.**

(A) Quantification of total fluorescence in single wild-type cells expressing Whi7-GFP (JCY2446) (*left panel*) or Whi5-GFP (JCY2448) (*right panel*) induced over night with the indicated concentrations of β-estradiol. GFP fluorescence and DIC images of representative cells are shown. (B) Quantification of total fluorescence in single *pho85* mutant cells expressing Whi7-GFP (JCY2561) or Whi5-GFP (JCY2559) induced overnight with the indicated concentrations of c-estradiol. Graph represents data from one experiment. Data information: In panels (A,B), boxes include 50% of data points, the line represents the median, and whiskers extend to maximum and minimum values. To normalize fluorescence intensity, value 1 was given to the mean of Whi7-GFP 0 nM. ns, $P > 0.05$, two-tailed unpaired *t*-test. $n$ = number of cells.

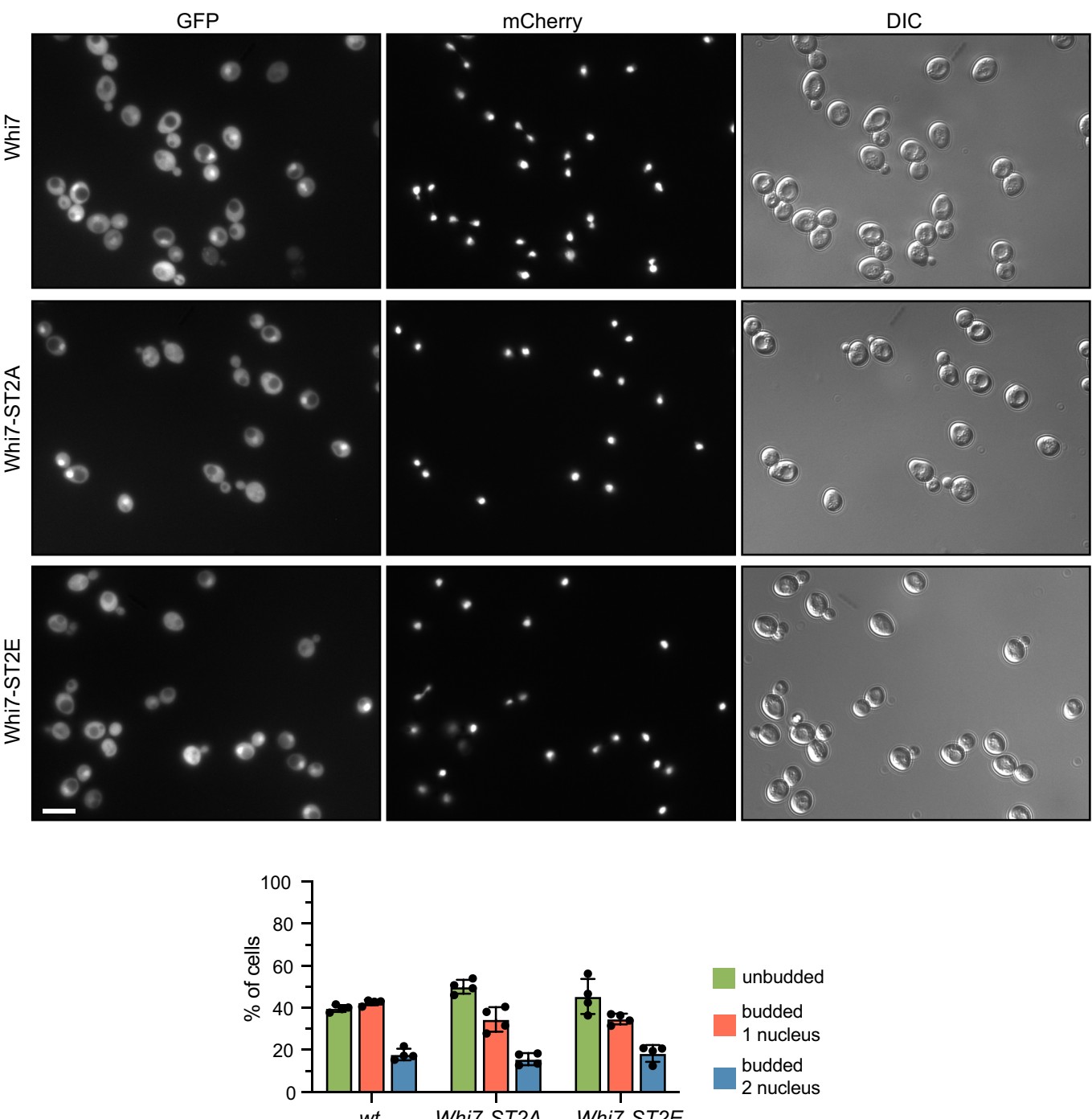

**Figure EV5. The subcellular localization of Whi7 is not affected by the Ser27 and Thr100 mutations.**

Exponentially growing cells of *whi7* cultures carrying Htb2-mCherry (JCY2981) transformed with a plasmid expressing *WHI7-4GFP*, *WHI7* [S27A, T100A]-*4GFP* or *WHI7* [S27E, T100E]-*4GFP* under the control of the *ADH1* promoter were imaged by fluorescence microscopy. GFP, mCherry and DIC channels are shown. Htb2-mCherry was used as nuclear marker. Scale bar 8 μm (upper panel). Cell cycle distribution of the imaged strains carrying the wild type and phosphomutant versions of Whi7 (*n* = 4). At least 200 cells were scored for each replicate and strain (lower panel). Data information: Data from the independent biological replicates are represented as mean ± s.d. *n* = number of biological replicates.

