## [Peer Review File · EMBO Reports]

The CDK Pho85 inhibits Whi7 Start repressor to promote cell cycle entry in budding yeast

Cristina Ros-Carrero, Mihai Spiridon-Bodi, J. Carlos Igual, and Mercè Gomar-Alba
DOI: 10.15252/embr.202357387

Corresponding author(s): Mercè Gomar-Alba (merce.gomar@uv.es), J. Carlos Igual (jcigual@uv.es)

Review Timeline:

Submission Date:	25th Apr 23
Editorial Decision:	1st Jun 23
Revision Received:	31st Oct 23
Editorial Decision:	7th Dec 23
Revision Received:	15th Dec 23
Accepted:	19th Dec 23

Editor: Deniz Senyilmaz Tiebe

Transaction Report:

Dear Dr. Gomar-Alba,

Thank you for the submission of your research manuscript for consideration by EMBO reports. It has now been seen by three experts in the field, and we have received the full set of their reports that are included below.

As you will see, the referees are generally positive about the study and acknowledge that the findings are potentially interesting and important for the yeast cell cycle and signaling fields. However, they also identify limitations in the study and raise a number of concerns that must be addressed. They all point out that the conclusions are not fully supported by the available data and, along these lines, referee #3 lists a number of important mechanistic questions that remain open. Although not all of those need to be addressed for further consideration of the manuscript, we agree with the referee that further exploration of the proposed mechanism at some of the important levels they summarize in their report would be desirable, as it would significantly strengthen the manuscript and substantiate the proposed mechanism. The referees also raise several technical and specificity concerns regarding the results (and their analysis and interpretation) of protein stability, cycloheximide chase, and in vitro kinase assays, which should be addressed, and they provide a number of detailed suggestions for the improvement of the study and the manuscript.

Given these constructive comments from the three well-informed reviewers, we would like to invite you to revise your manuscript with the understanding that their concerns (as detailed in their reports) must be fully addressed and their suggestions taken on board. Please address all referee concerns in a complete point-by-point response. Acceptance of the manuscript will depend on a positive outcome of a second round of review. It is EMBO reports policy to allow a single round of revision only and acceptance or rejection of the manuscript will therefore depend on the completeness of your responses included in the next, final version of the manuscript. If you have any questions or comments, we can also discuss the revisions in a video chat, if you like.

We realize that it is difficult to revise to a specific deadline. In the interest of protecting the conceptual advance provided by the work, we usually recommend a revision within 3 months (August 31st). Please discuss with me the revision progress ahead of this time if you require more time to complete the revisions.

IMPORTANT NOTE:

We perform an initial quality control of all revised manuscripts before re-review. Your manuscript will FAIL this control and the handling will be DELAYED if the following APPLIES:

- 1) If a data availability section providing access to data deposited in public databases is missing. If you have not deposited any data, please add a sentence to the data availability section that explains that (see below for more information).
- 2) If your manuscript contains statistics and error bars based on $n=2$. Please use scatter plots in these cases. No statistics should be calculated if $n=2$.

- 1) A .docx formatted version of the manuscript text (including legends for main figures, EV figures and tables). Please make sure that the changes are highlighted to be clearly visible.
- 2) Individual production quality figure files as .eps, .tif, .jpg (one file per figure). Please download our Figure Preparation Guidelines (figure preparation pdf) from our Author Guidelines pages <https://www.embopress.org/page/journal/14693178/authorguide> for more info on how to prepare your figures.
- 3) A .docx formatted letter INCLUDING the reviewers' reports and your detailed point-by-point responses to their comments. As part of the EMBO Press transparent editorial process, the point-by-point response is part of the Review Process File (RPF), which will be published alongside your paper unless you opt out of this (please see below for further information).
- 4) A complete author checklist, which you can download from our author guidelines (<<https://www.embopress.org/page/journal/14693178/authorguide>>). Please insert information in the checklist that is also reflected in the manuscript. The completed author checklist will also be part of the RPF (please see below for more information).

5) Please note that all corresponding authors are required to supply an ORCID ID for their name upon submission of a revised manuscript (<<https://orcid.org/>>). Please find instructions on how to link your ORCID ID to your account in our manuscript tracking system in our Author guidelines (<<https://www.embopress.org/page/journal/14693178/authorguide#authorshipguidelines>>)

6) We replaced Supplementary Information with Expanded View (EV) Figures and Tables that are collapsible/expandable online. A maximum of 5 EV Figures can be typeset. EV Figures should be cited as 'Figure EV1, Figure EV2' etc... in the text and their respective legends should be included in the main text after the legends of regular figures.

- For the figures that you do NOT wish to display as Expanded View figures, they should be bundled together with their legends in a single PDF file called *Appendix*, which should start with a short Table of Content (including page numbers). Appendix figures should be referred to in the main text as: "Appendix Figure S1, Appendix Figure S2" etc. See detailed instructions regarding expanded view here:
<<https://www.embopress.org/page/journal/14693178/authorguide#expandedview>>

7) Please note that a "Data availability" section at the end of Materials and Methods is now mandatory. In case you have no data that require deposition in a public database, please state so instead of refereeing to the database: "Our study includes no data deposited in public repositories." under the heading "Data availability". See also <<https://www.embopress.org/page/journal/14693178/authorguide#dataavailability>>. Please note that the Data availability statement is restricted to new primary data that are part of this study.

8) We request authors to consider both actual and perceived competing interests. Please review the new policy (<<https://www.embopress.org/competing-interests>>) and update your competing interests statement if necessary. Please name this section 'Disclosure and competing interests statement' and place it after the Acknowledgements section.

9) Figure legends and data quantification:
The following points must be specified in each figure legend:

- the name of the statistical test used to generate error bars and P values,
- the number (n) of independent experiments (please specify technical or biological replicates) underlying each data point,
- the nature of the bars and error bars (s.d., s.e.m.)
- If the data are obtained from n {less than or equal to} 2, use scatter plots showing the individual data points.

Discussion of statistical methodology can be reported in the Materials and Methods section, but figure legends should contain a basic description of n, P and the test applied.

See also the guidelines for figure legend preparation:
<https://www.embopress.org/page/journal/14693178/authorguide#figureformat>

10) We now request publication of original source data with the aim of making primary data more accessible and transparent to the reader. Our source data coordinator will contact you to discuss which figure panels we would need source data for and will also provide you with helpful tips on how to upload and organize the files.

11) Our journal encourages inclusion of *data citations in the reference list* to directly cite datasets that were re-used and obtained from public databases. Data citations in the article text are distinct from normal bibliographical citations and should directly link to the database records from which the data can be accessed. In the main text, data citations are formatted as follows: "Data ref: Smith et al, 2001" or "Data ref: NCBI Sequence Read Archive PRJNA342805, 2017". In the Reference list, data citations must be labeled with "[DATASET]". A data reference must provide the database name, accession number/identifiers and a resolvable link to the landing page from which the data can be accessed at the end of the reference. Further instructions are available at <<https://www.embopress.org/page/journal/14693178/authorguide#referencesformat>>.

12) Please also note our reference format:
<<http://www.embopress.org/page/journal/14693178/authorguide#referencesformat>>.

13) We now use CRediT to specify the contributions of each author in the journal submission system. CRediT replaces the author contribution section, which should be removed from the manuscript. Please use the free text box to provide more detailed descriptions. See also guide to authors:
<<https://www.embopress.org/page/journal/14693178/authorguide#authorshipguidelines>>.

14) As part of the EMBO publications' Transparent Editorial Process, EMBO reports publishes online a Review Process File to accompany accepted manuscripts. This File will be published in conjunction with your paper and will include the referee reports, your point-by-point response and all pertinent correspondence relating to the manuscript.

You can opt out of this by letting the editorial office know (emboreports@embo.org). If you do opt out, the Review Process File link will point to the following statement: "No Review Process File is available with this article, as the authors have chosen not to make the review process public in this case."

I look forward to seeing a revised version of your manuscript when it is ready. Please let me know if you have any questions or comments regarding the revision.

You can use this link to submit your revision: <https://embor.msubmit.net/cgi-bin/main.plex>

Yours sincerely,

Ioannis Papaioannou, PhD
Editor
EMBO reports

Referee #1:

In this paper, Ros-Carrero and co-workers investigate the role of the budding yeast cyclin-dependent kinase (CDK) Pho85 in regulating Whi7, a relatively recently identified repressor of the G1/S transition (Start). In doing so, the authors continue their investigation of Whi7, having already a recognizable publication track on the subject (Gomar-Alba et al., 2017, Méndez et al., 2020). Having previously established the differences between Whi7 and its more studied paralog Whi5, in this paper the authors show that Whi7 is regulated both at the transcriptional and at the post-translational level and show that Pho85 is involved in this regulation. The findings of this paper, as shown in the manuscript, are overall convincing and the conclusions are generally backed by the data. However, I think that issues with data presentation and some not entirely convincing results would need to be addressed by the authors (see below). The paper could be a nice addition to the fields of yeast cell cycle and signaling, challenging the simplistic view of an omnirelevant repressor (Whi5).

Major points:

1. Statistical analysis of the data is largely missing. Even when the authors show convincing data quantifications (and the relative single data points) data are not statistically evaluated. Statistics appear only in Figure 5. This is particularly needed when the authors make specific claims about somehow not so striking differences such as in Figure 2A, where they state that "...the inactivation of Pho80 and the combination of Pcl1, Pcl2, and Pcl9 mutations caused an additive increase in Whi7 protein levels". The effect of the mutations looks only mildly additive, and the claim should be properly supported by the statistics. On the same line: statistics are not mentioned neither in the Methods not in the Figure Legends.
2. Wherever the authors show protein stability assays they could calculate the protein's half-life (Figure 1D, 2C-D-E, 3C), especially when they make specific claims ("As shown in Figure 3C, abolishing the phosphorylation of Ser27 and Thr100 resulted in the stabilization of the Whi7 protein, with a Whi7 half-life similar to that observed upon Pho85 inactivation"). Also, for better understanding I would explicit in the blots labels the CHX treatment (as on the x-axis of graphs).
3. The use of cycloheximide to perform translation shut-offs might cause remarkable side-effects, for instance the quick and potent hyperactivation of TORC1, with unpredictable effects on the cell cycle proteome. Since the authors are evaluating the stability of a protein with a presumably short half-life they could validate their observations with transcription shut-offs.
4. The results of the in vitro kinase assays in Figure 3 are somehow not entirely convincing. Even upon incubation with the kinase, Whi7 is always largely present in the un-phosphorylated fast-migrating form. This would suggest that the kinase assay is not technically working very well (given the 30 minutes incubation stated in the Methods). The appearance of some slower migrating isoforms should at least be quantified to convince the reader of the goodness of such a weak signal. Also, reporting the microliters of kinase used is not very telling and the authors should make the effort of estimating the concentration.
5. The authors state that "This (WHI5 GAL1p induction) expression had no significant effect on cell proliferation (Figure 5C). However, the induction of Whi5 in wild type cells caused an accumulation of cells in the G1 phase (Figure 5D)". How could that be? Either the authors are hinting at a compensatory effect in G2/M or their measure of cell proliferation is not accurate. Either way, this should be discussed further. In Figure 5C the authors measure enough data points to properly calculate the doubling times of those strains.

6. I am not totally convinced of the approach used in Figure 5 to "equilibrate" Whi5/7 levels, notably because wt and pho85 Δ are treated separately. Considering that the authors use estradiol concentrations which are not enormously different (1 nM and 1.25 nM, for instance), is it not possible to find a compromise in order to better evaluate the results?

Minor points:

1. A sentence in the Abstract wrongly states that "...Pho85 up-regulates Whi7 protein levels through the control Whi7 protein stability and WHI7 gene transcription", where it should be "down-regulates".

2. In the Introduction, it is stated that "The budding yeast *S. cerevisiae* has six CDKs: Cdc28, Pho85, Kin28, Srb10, Bur1, and Ctk1.". Presumably the authors are referring to Ctk1 as "Ctk1". Should be clarified using a standard name.

3. Error bars in some graphs should be checked. While in most of the cases both the upper- and lower-whiskers are depicted, they are entirely lacking in Figure 2D (or are they too small?) and only the upper ones are shown in Figure 3C.

4. To strengthen the kinase-substrate relationship of Pho85 and Whi7, the authors might want to check their physical interaction (e.g. CoIP). Knowing that these interactions are very often transient and difficult to observe, they could use a kinase dead version of Pho85 (E53A, see Nishizawa et al, 1999). Using a KD mutant often helped in getting a stable interaction (it cannot phosphorylate and leave).

5. What is the explanation behind the odd migration pattern of Whi7 and Whi5, when loaded side-by-side (Figure 5B)? Provided that Whi7 is the faster migrating band (since Whi7 is smaller than Whi5) and that upper bands are its phosphoisoforms, why does Whi5 not show any phosphoisoform at all, since it is known to be a heavily phosphorylated protein (10+ sites)?

Referee #2:

Whi7 (Srl3) is the paralog of the well-studied G1/S inhibitor Whi5, and its role in controlling cell cycle entry is still poorly understood. In good lab growth conditions, deletion of Whi7 only leads to a weak phenotype, but recently it was shown that Whi7 becomes more important in stress conditions. In the present manuscript, Ros-Carrero et al. now identify Whi7 to be a target of Pho85. They identify Whi7 phosphosites targeted by Pho85, show that Whi7 is upregulated in pho85 deletion strains, and find corresponding phenotypes. They also find that Whi7 localization to Start gene promoters increases in the absence of Pho85. I found the manuscript interesting and think it contains important pieces of data. However, I am not yet fully convinced that the authors have sufficient evidence for the model they propose. This is largely due to some inconsistencies in the results and technical concerns, which would need to be addressed before publication.

Major concerns

1.) While the interpretation presented by the authors is certainly plausible, I am not entirely convinced that the interaction of Whi7 and Pho85 needs to be direct. An alternative explanation is that Pho85 deletion causes stress, which then leads to downstream Whi7 upregulation. In particular, I am concerned about the experiment described in the discussion, where the authors state that phosphate starvation does not lead to Whi7 overexpression. I do not fully understand the explanation presented by the authors. Would phosphate starvation not lead to an inactivation of Pho85? Should this then not lead to a dephosphorylation of Whi7 similar to the situation in the pho85 deletion? Given the potential importance of this experiment, I think the authors should show the corresponding data in the results section.

Along those lines, if Whi7 upregulation is dependent on Pho85 inactivation, does a pho85 deletion strain still show further overexpression of Whi7 in cell wall stress conditions?

2.) The quantification of Whi7 stability critically depends on linearity of the western blot quantification, however no controls are shown. This is particularly relevant given the different starting concentrations of Whi7 in the different mutants. For example, if the western blot quantifications is sublinear (which might be likely given the large range of concentrations quantified), the initially higher concentration of Whi7 in the pho85 deletion will lead to an apparently weaker decline upon translation inhibition. I think the only way to address this issue - besides rigorous control regarding linearity - is to load less proteins for the mutants that show higher Whi7 starting concentrations to achieve comparable bands at time 0.

3.) I assume the in vitro phosphorylation assays require a cyclin associated to Pho85, but I could not find this information in the manuscript.

4.) The authors claim that 'the Whi7S27A,T100A mutant was insensitive to the treatment with Pho85'. However, also the mutant shows clearly one (rather than two as in the WT) phosphobands at 120 and 150 ul, but not at 0 ul. Also, in contrast to what is claimed in the results, I did not find evidence for 'dose-dependent phosphorylation' since I do not see a difference between the 120 and 150 ul condition.

5.) In the in vivo experiments, e.g. Fig. 3g, I do not see an effect of pho85 deletion on the phosphobands of Whi7. How is that consistent with the authors' model?

6.) In Figure 3C, the authors present evidence for different stabilities of the Whi7 phosphomutants. Should this then not also lead to different protein concentrations, similar to the pho85 deletion situation? Looking at the western blots, the starting concentrations look quite similar... It would also be good to test also the pho85 deletion in the alanine mutant background (where no synergistic effect would be predicted according to the authors' model) and the phosphomimetic mutant in the PHO85 background.

7.) The experiment shown in Fig. 5 is interesting, and I am convinced that Whi7 becomes more important in the pho85 deletion background. However, I do not understand why Whi5 would become less important, i.e. why does Whi5 overexpression have hardly an effect in the deletion (Fig. 5H)?

8.) While not absolutely necessary for the claims the authors want to make, it would be nice to include quantifications of the nuclear localization of Whi5 and Whi7 discussed in Figure EV1. It could also be interesting to test whether this localization changes in the phosphomutants.

9.) Please include an explanation of how the fluorescent signals of the Whi7-GFP and Whi5-GFP have been quantified. While not necessary for the authors' claims, it could also be interesting to compare the nuclear concentrations.

Minor points

There are still some incomplete sentences in the manuscript. For example, last sentence of the introduction, and first section of the results: '... becoming the upregulation of Whi7 abolished....'.

Referee #3:

The manuscript titled "The CDK Pho85 inhibits Whi7 Start repressor to promote cell cycle entry in budding yeast" states that downregulation of protein kinase Pho85 activity up-regulates Whi7 protein levels through the control Whi7 protein stability and WHI7 gene transcription. Furthermore, they claim that the phosphorylation of two residues at Whi7 by Pho85 directly restrains the Whi7 association with promoters.

The findings are potentially interesting but still too preliminary. The authors do not explain the proposed mechanism at any of the crucial levels. First, how is the stability of Whi7 controlled by the phosphorylation of the two sites on Whi7? And for example, why one needs a larger concentration, the 10 nM dose, of estradiol in the absence of Pho85 when it should be more stable? Secondly, how do the two phosphorylated sites affect the binding of Whi7 to promoters and what role do the other CDK sites may have in this binding interaction? Is Whi7 phosphorylated at promoters or when not bound to promoters? No mass-spec analysis of the phosphorylation status is performed. Third, it is also unclear, for example, if Whi7 may enter the nucleus upon dephosphorylation. This possibility is not analyzed. Neither as an additional mechanism nor as an alternative to the promoter binding model (shifting equilibrium via concentration change in the nucleus). Fourth, the authors showed that the cyclins activating Pho85 play a redundant role in the unstabilization of Whi7. The protein kinase Pho85 has a pleiotropic role in signaling. It is a multifunctional CDK whose signaling tells the cell that the environmental conditions are favorable. The Pho85 deletion results in many defects such as slow growth in rich media, increased G1 duration, lethality in nonfermentable carbon sources, accumulation of glycogen, polarity defects and hypersensitivity to cell wall stress, osmotic stress, high concentrations of calcium or sodium, and DNA damage. So, do all these cyclins affect Whi7 phosphorylation? For example, how does a single signal downregulating a single cyclin activity affect this hypothetical Whi7 function if its role is redundant?

We thank all reviewers for their thorough assessment and constructive comments. For clarity, their comments have been numbered. Our replies are in blue. The introduced changes in the manuscript are marked in red.

Referee #1:

In this paper, Ros-Carrero and co-workers investigate the role of the budding yeast cyclin-dependent kinase (CDK) Pho85 in regulating Whi7, a relatively recently identified repressor of the G1/S transition (Start). In doing so, the authors continue their investigation of Whi7, having already a recognizable publication track on the subject (Gomar-Alba et al., 2017, Méndez et al., 2020). Having previously established the differences between Whi7 and its more studied paralog Whi5, in this paper the authors show that Whi7 is regulated both at the transcriptional and at the post-translational level and show that Pho85 is involved in this regulation. The findings of this paper, as shown in the manuscript, are overall convincing and the conclusions are generally backed by the data. However, I think that issues with data presentation and some not entirely convincing results would need to be addressed by the authors (see below). The paper could be a nice addition to the fields of yeast cell cycle and signaling, challenging the simplistic view of an omnirelevant repressor (Whi5).

Major points:

1. Statistical analysis of the data is largely missing. Even when the authors show convincing data quantifications (and the relative single data points) data are not statistically evaluated. Statistics appear only in Figure 5. This is particularly needed when the authors make specific claims about somehow not so striking differences such as in Figure 2A, where they state that "...the inactivation of Pho80 and the combination of Pcl1, Pcl2, and Pcl9 mutations caused an additive increase in Whi7 protein levels". The effect of the mutations looks only mildly additive, and the claim should be properly supported by the statistics. On the same line: statistics are not mentioned neither in the Methods not in the Figure Legends.

We have now added the statistics for all the Figures, mentioned the corresponding information in the Figure Legends and included a *Statistical Methods and Reproducibility* section in Materials and Methods.

Data from shut offs and Whi7 protein and mRNA levels in Figure 2 supports the collective contribution of *Pcl1,2,9* and *Pho80* cyclins in the control of Whi7 protein levels, suggesting an accumulative effect of the four mutations. However, we agree with the reviewer that differences between *pho80* and *pcl1,2,9,pho80* in Figure 2A are not statistically significant (two-tailed t-test p-value = 0.51). We have changed the text to make this point clearer (p7).

2. Wherever the authors show protein stability assays they could calculate the protein's half-life (Figure 1D, 2C-D-E, 3C), especially when they make specific claims ("As shown in Figure 3C, abolishing the phosphorylation of Ser27 and Thr100 resulted in the stabilization of the Whi7 protein, with a Whi7 half-life similar to that observed upon Pho85 inactivation"). Also,

for better understanding I would explicit in the blots labels the CHX treatment (as on the x-axis of graphs).

We thank the reviewer for pointing this out. We have calculated the degradation rate constant (k_d) extracted from the slope of the linear regression in all the translational shut-offs. Comparison of the k_d of the mutants showing different Whi7 protein stabilities are now shown as bar graphs in the Figures with the corresponding statistics.

In Figure 3C the mean of k_d is 0.0070 and 0.0108 for *pho85* and *Whi7-2A* respectively, and the p-value = 0.072 (ns). We have changed the text and refer to k_d instead of protein half-life “...abolishing the phosphorylation of Ser27 and Thr100 resulted in the stabilization of the *Whi7* protein, with a *Whi7 kd* significantly reduced compared to that of wild type protein and very similar to that observed upon *Pho85* inactivation (Figure 3C)”.

Note that *Whi7 kd* in the experimental approach of Figure 3 and Figure EV3 (sc-trp media and plasmid) is lower than that observed in YPD media and genomic-tagged strains (Figure 1, Figure 2). This is commented now in the main text (p9).

We have included the CHX label in the western blots of Figures 1, 2 and 3, as suggested.

3. The use of cycloheximide to perform translation shut-offs might cause remarkable side-effects, for instance the quick and potent hyperactivation of TORC1, with unpredictable effects on the cell cycle proteome. Since the authors are evaluating the stability of a protein with a presumably short half-life they could validate their observations with transcription shut-offs.

Cycloheximide (CHX) has been extensively used to investigate protein stability, due to its role inhibiting the last stage of the protein synthesis. In principle, any CHX-derived effect might be affecting both wild type and *pho85* cells. We do not think CHX may act causing artefactual stabilization of cell-cycle regulators specifically in *pho85*, since it has been also used in other works to investigate other *Pho85* targets. For example, the G1 cyclin *Cln3*, a short-lived protein whose stability is decreased in the *pho85* mutant (Menoyo et al., *MolCellBiol* 2013, PMID: 23339867, Figure 5B). Thus, using CHX, *Cln3* half-life in the absence of the CDK showed the opposite effect than *Whi7*. We also assayed the stability of the mitotic cyclin *Cib2* using the same assays, that resulted unaffected by *Pho85* (Reviewer Figure 1A).

Reviewer Figure 1. A. *Cib2* protein stability was analyzed by translational shut-off with cycloheximide (CHX). Wild type and *pho85* cells were incubated in the presence of CHX 100 μ g/mL and *Cib2*-HA protein level was analyzed at the indicated times after the addition of CHX by western blot. **B.** *Whi7* protein stability was analyzed by transcriptional shut-off with doxycycline. Wild type and *pho85* cells expressing TetO:WHI7-GFP were grown in the presence of doxycycline 10 μ g/mL and then *WHI7*

expression was induced by removing doxycycline for 12h. Doxycycline 10µg/mL was added to exponential cells to block *WHI7* transcription and Whi7-GFP protein level was analyzed at the indicated times by western blot. **A-B.**

Furthermore, we have performed transcriptional shut-off with doxycycline to block TetO:WHI7-GFP expression and we also observe increased Whi7 protein stability in *pho85* mutant (Reviewer Figure 1B). Taken together, these results demonstrate that hyperactivation of TORC1 and/or other CHX side effects do not invalidate the main conclusions of our protein stability assays.

4. The results of the *in vitro* kinase assays in Figure 3 are somehow not entirely convincing. Even upon incubation with the kinase, Whi7 is always largely present in the unphosphorylated fast-migrating form. This would suggest that the kinase assay is not technically working very well (given the 30 minutes incubation stated in the Methods). The appearance of some slower migrating isoforms should at least be quantified to convince the reader of the goodness of such a weak signal. Also, reporting the microliters of kinase used is not very telling and the authors should make the effort of estimating the concentration.

We have technically improved the conditions of the *in vitro* kinase assay. Now we have obtained unambiguous slow-migrating bands and quantified the % of these phosphorylated isoforms from three independent experiments. We think new Figure 3B convincingly shows that Pho85 phosphorylates Whi7 *in vitro* and that this phosphorylation is significantly reduced in the Whi7-ST2A mutant.

We agree with the reviewer that labelling the blot with the microliters of kinase used in the reaction is not experimentally informative. The levels of a single protein obtained from a mini (2-4L of initial culture) TAP elution are not routinely quantified due to its low concentration and the presence of co-interacting proteins and/or IgGs that can mask the "real" kinase concentration. Instead, we referred to the initial number of cells used for the TAP purification in *Materials and Methods* section. In any case, we would like to remark that in our assay (Figure 3B) the same kinase amount is used for the wt and 2A mutant.

5. The authors state that "This (WHI5 GAL1p induction) expression had no significant effect on cell proliferation (Figure 5C). However, the induction of Whi5 in wild type cells caused an accumulation of cells in the G1 phase (Figure 5D)". How could that be? Either the authors are hinting at a compensatory effect in G2/M or their measure of cell proliferation is not accurate. Either way, this should be discussed further. In Figure 5C the authors measure enough data points to properly calculate the doubling times of those strains.

Some mutants of G1/S repressors (*whi3*, *sic1*, *hos3*), show reduced G1 duration (and reduced cell size) without presenting changes in their doubling times (see for example Soifer and Barkai, *Molecular Systems Biology*, 2014 PMID: 25411401; Kumar et al., *NCB* 2018 PMID: 29531309). Indeed, this suggests compensatory mechanisms for the other cell cycle phases, although the underlying mechanisms are not completely understood. We now mention this in the main text in order to clarify this point (p11).

We determine cell number with a Coulter counter, which in our experience is a highly accurate methodology that provides reliable and consistent data. We have now added this information in the *Materials and Methods* section.

In our opinion, given the similar profiles obtained in Figure 5C it is not informative to indicate the doubling times and we do not expect significant differences.

6. I am not totally convinced of the approach used in Figure 5 to "equilibrate" *Whi5/7* levels, notably because *wt* and *pho85Δ* are treated separately. Considering that the authors use estradiol concentrations which are not enormously different (1 nM and 1.25 nM, for instance), is it not possible to find a compromise in order to better evaluate the results?

We understand the point of the Reviewer that it would have been more intuitive to use the same concentrations for both backgrounds in the Figure. However, in Figure 5, we seek to compare similar protein levels of *Whi7* and *Whi5* independently in the two backgrounds: *wt* and *pho85*. We first thoroughly screened for the concentrations of β -estradiol that allowed us to have similar levels of *Whi5* and *Whi7* in a *wt* strains (Figure EV4A). Because there is a quick increase of the *Whi5* induction between 1nM and 5nM β -estradiol, for *pho85* we checked the concentrations shown in Figure EV4B and 1.25nM but not 1nM results in similar levels than *Whi7* 10nM. In addition, when we induce in a *wt* background *Whi7* 10 nM and *Whi5* 1.25 nM, there is more *Whi7* protein than *Whi5* (Reviewer Figure 2), suggesting that the concentrations that work for one background may not be exactly the same that work for the other background.

Reviewer Figure 2. Protein levels of *Whi7*-GFP and *Whi5*-GFP determined by western blot in the wild type cells. *Whi7*-GFP and *Whi5*-GFP were induced with 10 nM and 1.25 nM of β -estradiol respectively.

However, we would like to point out that we do not pretend to have exactly the same levels of *Whi7* in *wt* (7.5nM) and *pho85* (10nM) backgrounds, or the same levels of *Whi5* in *wt* (1nM) and *pho85* (1.25). In this Figure we understand each background as an independent experiment, although we were very careful to coherently move between similar levels of induction in both backgrounds. In this sense, we also would like to remark that we use a slightly higher dose in *pho85* mutant based on the fact that the same dose of β -estradiol always results in a lower gene induction in a *pho85* background when compared to a *wt* strain (Reviewer Figure 3).

Reviewer Figure 3. Wild type and *pho85* cells expressing *GAL1:WHI7-GFP* were treated for 1h with β -estradiol 1nM. Total fluorescence intensity of *Whi7*-GFP signal was measured in single cells by segmentation of whole-cell in the DIC channel. To normalize fluorescence

intensity, value 1 was given to the mean of Whi7-GFP in wild type strain. Quantification corresponds to cells shown in Figure EV1B.

Minor points:

1. A sentence in the Abstract wrongly states that "...Pho85 up-regulates Whi7 protein levels through the control Whi7 protein stability and WHI7 gene transcription", where it should be "down-regulates".

We have corrected this.

2. In the Introduction, it is stated that "The budding yeast *S. cerevisiae* has six CDKs: Cdc28, Pho85, Kin28, Srb10, Bur1, and Ctrk1.". Presumably the authors are referring to Ctk1 as "Ctrk1". Should be clarified using a standard name.

We have corrected this.

3. Error bars in some graphs should be checked. While in most of the cases both the upper- and lower-whiskers are depicted, they are entirely lacking in Figure 2D (or are they too small?) and only the upper ones are shown in Figure 3C.

This has been fixed. We now show Figures 2D and 3C with complete error bars.

4. To strengthen the kinase-substrate relationship of Pho85 and Whi7, the authors might want to check their physical interaction (e.g. CoIP). Knowing that these interactions are very often transient and difficult to observe, they could use a kinase dead version of Pho85 (E53A, see Nishizawa et al, 1999). Using a KD mutant often helped in getting a stable interaction (it cannot phosphorylate and leave).

We thank the reviewer for suggesting this experiment. We now show in new Figure 3A that the kinase dead version of Pho85 indeed physically interacts with Whi7 by CoIP. We were not able to detect interaction between Whi7 and the wild type version of Pho85, probably due to the transient interactions between kinases and their targets, as the Reviewer points out. This result reinforces the conclusion that Whi7 is an *in vivo* target of Pho85.

Moreover, we carried out phospho-proteomics in wt and *pho85* cells, now included in the revised version of the manuscript, which supports that the inactivation of Pho85 *in vivo* affects the phosphorylation state of some CDK consensus sites of Whi7 (Table 1 and Appendix Figure S1), further supporting that Whi7 is a direct target of Pho85.

5. What is the explanation behind the odd migration pattern of Whi7 and Whi5, when loaded side-by-side (Figure 5B)? Provided that Whi7 is the faster migrating band (since Whi7 is smaller than Whi5) and that upper bands are its phosphoisoforms, why does Whi5 not show

any phosphoisoform at all, since it is known to be a heavily phosphorylated protein (10+ sites)?

Whi5 has been extensively detected by western blot by many labs and, unlike Whi7, the Whi5 phosphorylation pattern cannot be clearly observed in conventional SDS-PAGE electrophoresis and western blot. Using routine SDS-PAGE gels, in the best of cases, PhosphoWhi5 can be distinguished as a shadow over the major band (Figure 5F and references below). Therefore, Phos-tag gels have been oftenly used to clearly distinguish and track Whi5 phosphoisoforms. See for example blots in Gomar-Alba, et al., Nat Comm, 2017 (Figure 7) PMID: 28839131; Qu, et al., Cell Reports, 2019 (Figure 2) PMID: 31644918; Kõivomägi, et al., Science, 2021 (Figure 1) PMID: 34648313, among others. In the case of Whi7, we and others (Yahya et al., MolCell 2014, PMID: 24374311) detect multiple low migration bands in regular SDS-PAGE gels.

Referee #2:

Whi7 (Srl3) is the paralog of the well-studied G1/S inhibitor Whi5, and its role in controlling cell cycle entry is still poorly understood. In good lab growth conditions, deletion of Whi7 only leads to a weak phenotype, but recently it was shown that Whi7 becomes more important in stress conditions. In the present manuscript, Ros-Carrero et al. now identify Whi7 to be a target of Pho85. They identify Whi7 phosphosites targeted by Pho85, show that Whi7 is upregulated in *pho85* deletion strains, and find corresponding phenotypes. They also find that Whi7 localization to Start gene promoters increases in the absence of Pho85. I found the manuscript interesting and think it contains important pieces of data. However, I am not yet fully convinced that the authors have sufficient evidence for the model they propose. This is largely due to some inconsistencies in the results and technical concerns, which would need to be addressed before publication.

Major concerns

1.) While the interpretation presented by the authors is certainly plausible, I am not entirely convinced that the interaction of Whi7 and Pho85 needs to be direct. An alternative explanation is that Pho85 deletion causes stress, which then leads to downstream Whi7 upregulation.

Following Reviewer 1 suggestion, we have performed CoIP assays between Pho85 and Whi7 to prove their direct physical interaction. In new Figure 3A, the immunoprecipitated Whi7 interacts with the catalytically dead version of Pho85 (Pho85-E53A-HA, Wanke et al., *The EMBO Journal* (2005) 24:4271-4278). This result, together with other observations in the manuscript, clearly supports a direct regulation of Pho85 to Whi7. Furthermore, we have now included Whi7 phosphoproteomic analysis in wt and *pho85* cells, which supports that the inactivation of Pho85 *in vivo* affects the phosphorylation state of some CDK consensus sites of Whi7 (Table 1 and Appendix Figure S1), further supporting that Whi7 is a direct target of Pho85.

In particular, I am concerned about the experiment described in the discussion, where the authors state that phosphate starvation does not lead to Whi7 overexpression. I do not fully understand the explanation presented by the authors. Would phosphate starvation not lead to an inactivation of Pho85? Should this then not lead to a dephosphorylation of Whi7 similar to the situation in the *pho85* deletion? Given the potential importance of this experiment, I think the authors should show the corresponding data in the results section.

In general terms, phosphate limitation causes inactivation of Pho85. However, in this condition, the CKI Pho81 specifically inhibits Pho85-Pho80 and Pho85-Pcl7 signalling (Lee et al., *Nat Chem Biol.* 2008. PMID: 18059263; Lee et al., *Molecular Microbiology.* 2000. PMID: 11069666), which in turn triggers the activation of the Pho4 transcription factor. Our results show that there is no increase in mRNA levels nor Whi7 protein stability under phosphate starvation (Figure EV2). We do not see any inconsistency on this. No Pho4 consensus binding sites are found in sequence analysis of the *WHI7* promoter and indeed we could not detect association of Pho4 to *WHI7* promoter by ChIPs assays (Figure EV2A). Therefore, the increase of Whi7 mRNA observed in *pho80* and *pho85* mutants (Figure 2B)

may be explained by the activation of other(s) transcription factors. Additionally, the deletion of *PHO80* could have a more severe effect on *WHI7* transcription than the inhibited Pho80-Pho85 activity caused by phosphate limitation. Regarding *Whi7* protein stability, note that *pho80* deletion has a limited effect in *Whi7 kd* (Figure 2D). We hypothesize that *Whi7* protein stability remains roughly constant in phosphate depletion probably because the CDK can be still acting in combination with Pcl1/2/9 cyclins that down-regulate *Whi7* levels. Overall, this may mask/compensate the effect of Pho80-Pho85 inactivation. As the reviewer suggested, we now show all these results in Figure EV2.

Along those lines, if *Whi7* upregulation is dependent on Pho85 inactivation, does a *pho85* deletion strain still show further overexpression of *Whi7* in cell wall stress conditions?

We do not see in *pho85* additive induction in *Whi7* protein levels in cell wall stress caused by Calcofluor White (Reviewer Figure 4). However, addressing this question (Pho85-dependent *Whi7* regulation in stress) may need a deeper analysis (for example, the dynamics of stress response in wt and *pho85* cells are probably different). As we comment in the discussion (p16), this is beyond the scope of this paper, but we would like to explore the question in future studies.

Reviewer Figure 4. *Whi7*-HA protein levels were analyzed by western blot in wild type and *pho85* cells treated with Calcofluor White (CWF) 10 μ g/mL at the indicated times. Graph represents the mean of two independent biological replicates.

2.) The quantification of *Whi7* stability critically depends on linearity of the western blot quantification, however no controls are shown. This is particularly relevant given the different starting concentrations of *Whi7* in the different mutants. For example, if the western blot quantifications is sublinear (which might be likely given the large range of concentrations quantified), the initially higher concentration of *Whi7* in the *pho85* deletion will lead to an apparently weaker decline upon translation inhibition. I think the only way to address this issue - besides rigorous control regarding linearity - is to load less proteins for the mutants that show higher *Whi7* starting concentrations to achieve comparable bands at time 0.

We are aware of the problem that differences in band intensity could cause in the quantification of proteins, and because of that we were very careful on this item. We do quantify expositions with similar intensities. Moreover, to estimate the linear range of the method, we carried out control quantification in low and high exposed membranes to check that quantification was linear in all the range of detected signal intensities.

For example, Reviewer Figure 5 shows the western blot of Figure 1D. From Pho85 t0 to t80 (higher and lower signal, respectively) the quantification is reliable, and the wt shows the same linearity in the upper panel (low exposure) than in the bottom panel (high exposure).

We have now included a sentence in the Methods section clarifying this point.

Reviewer Figure 5. The wild type strain of the translational shut-off shown in Figure 1D was quantified in western blot membranes with different exposures. Note that in the higher exposure, t0 of wild type and *pho85* strains show similar intensities.

3.) I assume the *in vitro* phosphorylation assays require a cyclin associated to Pho85, but I could not find this information in the manuscript.

For the *in vitro* kinase assay, Pho85 kinase tagged with TAP was isolated from yeast cells using Tandem Affinity Purification (TAP). Thus, any Pho85 co-interacting cyclin can be present in the assay. We described the technical details of the TAP purification and the kinase assay in the Materials and Methods section.

4.) The authors claim that 'the Whi7S27A,T100A mutant was insensitive to the treatment with Pho85'. However, also the mutant shows clearly one (rather than two as in the WT) phosphobands at 120 and 150 ul, but not at 0 ul. Also, in contrast to what is claimed in the results, I did not find evidence for 'dose-dependent phosphorylation' since I do not see a difference between the 120 and 150 ul condition.

We agree. We have technically improved kinase assay maximizing the amount of kinase (see our reply to Reviewer 1, point 4) and we have quantified the phosphobands (% of slow-migrating isoforms) in Whi7 and Whi7-ST2A from three independent biological replicates. As shown in new Figure 3B, Whi7-ST2A reduces Whi7 phosphorylation by approximately half in the *in vitro* kinase assay. This is consistent with the proteomic data (Table 1, Appendix Figure S1), where we have shown that in addition of S27 and T100, Pho85 could also mediate phosphorylation of other Whi7 residues. We have accordingly modified the main text (p8-9).

5.) In the *in vivo* experiments, e.g. Fig. 3g, I do not see an effect of *pho85* deletion on the phosphobands of Whi7. How is that consistent with the authors' model?

Mass-spectrometry analyses revealed up to 24 phosphosites in Whi7, and showed that Whi7 is still hyperphosphorylated in the absence of Pho85, which could explain the presence of abundant slow migrating bands in the mutant. Moreover, it can be expected a high redundancy and/or interplay between Cdc28 and Pho85 for phosphorylating the 12 CDK consensus sites of Whi7. Therefore, using conventional SDS-PAGE electrophoresis, we do not expect qualitative differences in Whi7 phosphopattern between wt and *pho85* cells. We have added a sentence in the main text to clarify this point (p6).

6.) In Figure 3C, the authors present evidence for different stabilities of the Whi7 phosphomutants. Should this then not also lead to different protein concentrations, similar to the *pho85* deletion situation? Looking at the western blots, the starting concentrations look quite similar... It would also be good to test also the *pho85* deletion in the alanine mutant background (where no synergistic effect would be predicted according to the authors' model) and the phosphomimetic mutant in the PHO85 background.

Figure 3C now includes degradation constant (*kd*) and statistics to better compare the different strains. In addition, we now show in Figure 3D the protein amount of Whi7-ST2A and *pho85* Whi7-ST2E, which includes times 0 from the shut-offs shown in Figure 3C. Note that in the western blots of Figure 3C panels from the different strains are separated, and different westerns or blot exposures are shown so they are not directly comparable.

Finally, we have now analyzed Whi7 stability in the additional mutants proposed by the Reviewer (Figure EV3). As it can be seen, there is no additive effect of *pho85* inactivation on Whi7-ST2A or Whi7-ST2E. This reinforces the conclusion that, at least in the control of protein stability, Pho85 is acting mainly through these two residues.

7.) The experiment shown in Fig. 5 is interesting, and I am convinced that Whi7 becomes more important in the *pho85* deletion background. However, I do not understand why Whi5 would become less important, i.e. why does Whi5 overexpression have hardly an effect in the deletion (Fig. 5H)?

As described in the manuscript, the *pho85* mutant shows a huge variety of phenotypes, including slow growth, increased G1 duration and stress sensibility. In Figure 5D (wt) and Figure 5H (*pho85*) we decided to quantify budding index distribution at 4h after β -estradiol addition, because at this timepoint *pho85* cells overexpressing Whi7 had already arrested proliferation (Figure 5G). We now added in Figure 5H the budding index quantifications corresponding to 6h after β -estradiol addition. At 6h, the induction of Whi5 in the absence of Pho85 shows an accumulation of G1 cells similar to that observed in the wild type strain at 4h (aprox. 50% of unbudded cells). We also obtained similar results at 8h. We conclude that although the β -estradiol-dependent gene induction and/or growing kinetics in wt and *pho85* are probably slightly different, Whi5 overexpression has a similar effect in both backgrounds.

8.) While not absolutely necessary for the claims the authors want to make, it would be nice to include quantifications of the nuclear localization of Whi5 and Whi7 discussed in Figure EV1. It could also be interesting to test whether this localization changes in the phosphomutants.

We agree. We have now added for the microscopy images the quantification of the % of cells in the different cell cycle phases, the % of cells with nuclear Whi7 in the different cell cycle phases, and the single cell quantification of the nuclear Whi7 amount in G1 cells.

For the nuclear quantifications, we have used the Htb2-mCherry signal to segment the nucleus. Whi7-NG allows to qualitatively analyze Whi7 localization at endogenous level, which, as can be observed in Figure EV1A, does not change in the *pho85* mutant. However, as we previously described (Méndez et al., 2020, PMID: 33443080) endogenous Whi7-NG signals are very weak and nuclear localization is sometimes difficult to visualize. Thus, to obtain robust nuclear quantifications we used Whi7 mildly induced with β -estradiol (Figure EV1 and Figure 5) or pADH:Whi7-4GFP plasmids to analyze the effect of S27 T100 mutations (Figure 7, Figure EV5).

All data support the conclusion that Pho85 does not significantly affect Whi7 subcellular localization.

Note that there is a slight increase in the percentage of G1 cells with nuclear signal in the *pho85* mutant (Figure EV1C). However, this is probably related to the slight increase in the % of G1 cells in *pho85* (EV1B) as a consequence of Whi7 induction (as occurs in Figure 7) that might reflect a slightly longer G1, which could facilitate the detection of nuclear Whi7. Indeed, *pho85* Whi7-NG do not show this G1 accumulation (Figure EV1A).

9.) Please include an explanation of how the fluorescent signals of the Whi7-GFP and Whi5-GFP have been quantified. While not necessary for the authors' claims, it could also be interesting to compare the nuclear concentrations.

We have added in Figure 5 the quantification of total Whi7 and Whi5 amount (whole cell) and nuclear Whi7/5 amount, in the *pho85* cells expressing the same levels of both repressors. As shown in Figure 5I, G1 cells also show the same total amount of both repressors, although Whi5 presents a higher nuclear amount than Whi7. This result further strengthens our conclusion, because despite having more nuclear amounts of Whi5 in G1 cells, at this same time point (4h), Whi7 induction (but not Whi5) causes a G1 blockage (Figure 5H).

We have completed the information of the quantification of fluorescent signals in the Materials and Methods section.

Minor points

There are still some incomplete sentences in the manuscript. For example, last sentence of the introduction, and first section of the results: '... becoming the upregulation of Whi7 abolished....'.

We have modified these sentences.

Referee #3:

The manuscript titled "The CDK Pho85 inhibits Whi7 Start repressor to promote cell cycle entry in budding yeast" states that downregulation of protein kinase Pho85 activity up-regulates Whi7 protein levels through the control Whi7 protein stability and WHI7 gene transcription. Furthermore, they claim that the phosphorylation of two residues at Whi7 by Pho85 directly restrains the Whi7 association with promoters.

1. The findings are potentially interesting but still too preliminary. The authors do not explain the proposed mechanism at any of the crucial levels. First, how is the stability of Whi7 controlled by the phosphorylation of the two sites on Whi7?

We have previously characterized that Whi7 is degraded via SCF^{Grr1} ubiquitin ligase complex in a phosphorylation dependent way (Gomar-Alba et al., 2017). We hypothesized that the phosphorylation of S27 and T100 in Whi7 by Pho85 promotes Whi7 instability by targeting the repressor for degradation via SCF^{Grr1}. Now we have checked this hypothesis and, indeed, the physical interaction between Whi7 and Grr1ΔF (lacks the F-box domain to bind substrate without degrading it) is reduced in the Whi7-ST2A phospho-null mutant (New Figure 3E).

And for example, why one needs a larger concentration, the 10 nM dose, of estradiol in the absence of Pho85 when it should be more stable?

β-estradiol induction dynamics is different in wt and *pho85* backgrounds. Therefore, the induction of *GAL1pr:WHI7* (using the same β-estradiol concentration and time) results in lower Whi7 levels in *pho85* than in wt cells (see Reviewer Figure 3 and Answer to Reviewer 1 point 6). Protein levels are a balance between transcription and degradation, and in this case the increased stability in *pho85* is not enough to compensate for the less efficient transcriptional induction with β-estradiol. Therefore, in Figure 5 we use slightly higher concentration for *pho85* than for the wt.

2. Secondly, how do the two phosphorylated sites affect the binding of Whi7 to promoters and what role do the other CDK sites may have in this binding interaction?

Whi7 has 12 CDK consensus sites. We think there is redundancy and/or interplay between Cdc28 and Pho85 for these phospho-sites, as commented now in p6. The mutation of these 12 CDK sites to Ala (Whi7-NP) results in a constitutively nuclear Whi7 and an increased binding to Start promoters (Gomar-Alba et al., 2017, PMID: 28839131; Méndez et al., 2020, PMID: 33443080). These observations indicate that Whi7 function is restricted by phosphorylation, a common trait of Start transcriptional repressors like Whi5 in yeast or Rb in mammals. In the original version of the manuscript, we described that the mutation of the two Pho85 strict consensus sites, S27 and T100 to Glu (Whi7-ST2E) reduced Whi7 binding to promoters. Now we have demonstrated that this effect is not due to changes in nuclear localization nor nuclear levels of the protein (Figure 7E-F, Figure EV5) but to a reduced affinity to Swi4 (New Figure 7G). Therefore, we conclude that the phosphorylation of these two sites by Pho85 inhibits the association of Whi7 to promoters through impairing Whi7 interaction with the SBF subunit Swi4.

Is Whi7 phosphorylated at promoters or when not bound to promoters?

We find this point difficult to address with conclusive experiments. In fact, although it is a well-known model that Cdc28-Cln phosphorylates Whi5 to dissociate it from SBF and export it out from the nucleus, to our knowledge it has not been proved that this phosphorylation occurs only when Whi5 is bound to promoters. According to our unpublished data, Whi7 phosphorylation can occur when Whi7 is anchored to the plasma membrane (using FRB-FKBP Rapamycin system) and also when Whi7 is constitutively nuclear (Whi7-NLS). This phosphorylation can be carried out by either Cdc28, Pho85 or any of the putative kinases that acts on the non-CDK sites detected in our phospho-proteomics assay (Table 1, Appendix Table S1). Furthermore, we also tried to investigate Whi7-WIQ (with the GTB domain mutated), which would have allowed to investigate Whi7 out from promoters, but, in our hands, it is still able to bind to promoters (Unlike Whi5-WIQ, Travesa et al., MolCellBiol 2013, PMID: 23382076). These are the obvious approaches to this question. We think going further in this point is beyond the scope of the manuscript at this moment.

No mass-spec analysis of the phosphorylation status is performed.

We have carried out TiO₂ phosphopeptide enrichment and tandem mass spectrometry (LC-MS/MS) using purified Whi7-GFP. The analyses of Whi7 phosphopeptides revealed the existence of 24 phosphorylation sites in wild type cells (Table 1, Appendix Table S1), detected in at least 2 of the 3 biological replicates and applying an astringent threshold of best localization probability >0.85. Note that 11 of these sites were CDK consensus sites (Cdc28 or Pho85). In the *pho85* mutant, we found 17 phosphorylated sites, including 8 CDK consensus sites. In addition to the loss of three CDK sites, when we compare the relative intensities of the CDK-consensus sites referred to the non-CDK sites in each sample, several of the CDK-consensus sites, including S27 and T100, are relatively less phosphorylated in the *pho85* mutant (Appendix Figure S1). This result supports that Whi7 is an *in vivo* target of Pho85, and suggests a redundancy of Cdc28 and Pho85 in the phosphorylation of many CDK consensus site. However, it is necessary to be cautious with the *pho85* results, since optimal Whi7 phosphopeptides detection required Whi7 mild overexpression that has slight cell cycle effects. In any case, we think that this proteomic data, together with the Pho85-Whi7 CoIP (New Figure 3A) and the new *in vitro* kinase assay (New Figure 3B) proves that Whi7 is a *bona fide* target of Pho85.

Peptides and phosphopeptides from Whi7/Srl3 identified in each sample are listed in Appendix_Dataset_S1 and Appendix_Dataset_S2.

Mass-spectrometry Data are available via ProteomeXchange via PRIDE with identifier PXD046448, accessible for Reviewers with the following details:

Username: reviewer_pxd046448@ebi.ac.uk **Password:** B9jMPrMR

3. Third, it is also unclear, for example, if Whi7 may enter the nucleus upon dephosphorylation. This possibility is not analyzed. Neither as an additional mechanism nor as an alternative to the promoter binding model (shifting equilibrium via concentration change in the nucleus).

A complete set of microscopy experiments to analyze Whi7 nuclear distribution are now shown in Figure EV1, Figure 5, Figure EV5 and Figure 7. See also our response to Reviewer 2 points 8-9. We have added Htb2-mCherry to segment the nuclear signal and we also use it as control of nuclear levels.

In the revised version of the manuscript we now show:

-Whi7 nuclear distribution along cell-cycle phases is not affected in *pho85* or in the Whi7-ST2A and Whi7-ST2E mutants

-Quantification of % of nuclear Whi7 protein in G1 single cells do not show differences between wt and *pho85* or between Whi7wt and Whi7-ST2A and Whi7-ST2E

-In *pho85* background, when we induce Whi7 and Whi5 to obtain the same protein levels, the quantification of G1 single cells revealed that Whi7 and Whi5 present equal total amounts (whole cell) but increased nuclear Whi5 amount; in these same cells, Whi7 induction (but not Whi5) causes G1 arrest in spite of having less nuclear protein, further confirming our results that Whi7 plays a more important role than Whi5 in the absence of Pho85.

-All these microscopy results support and strengthen our model and we have accordingly commented them in the *Discussion* section.

Details about the fluorescent signal quantifications are now extended in the *Materials and Methods* section, and in the corresponding Figure Legends.

4. Fourth, the authors showed that the cyclins activating Pho85 play a redundant role in the destabilization of Whi7. The protein kinase Pho85 has a pleiotropic role in signaling. It is a multifunctional CDK whose signaling tells the cell that the environmental conditions are favorable. The Pho85 deletion results in many defects such as slow growth in rich media, increased G1 duration, lethality in nonfermentable carbon sources, accumulation of glycogen, polarity defects and hypersensitivity to cell wall stress, osmotic stress, high concentrations of calcium or sodium, and DNA damage. So, do all these cyclins affect Whi7 phosphorylation? For example, how does a single signal downregulating a single cyclin activity affect this hypothetical Whi7 function if its role is redundant?

Functional redundancy between cyclins is a common trait in the regulation of the molecular effectors of the cell cycle. Because the ultimate goal of the Pho85-cyclin signalling on Whi7 is to promote Start activation through the downregulation of the repressor, we assayed which cyclins are involved in this G1 control. With this purpose, we followed the same experimental setup shown in Figure 4. As reviewer suggested, we assayed single *pho80*, *pcl1*, *pcl2*, and *pcl9* mutants, and we found that only in the absence of Pcl1 the overexpression of Whi7 caused G1 arrest (New Figure 4F-G and Appendix Figure S2). This result clearly demonstrates that Pho85-Pcl1 signalling has a major role in the inhibition of Whi7 function as a Start repressor.

Dear Dr. Gomar-Alba,

Thank you for submitting your revised manuscript. It has now been seen by all of the original referees. As my colleague Ioannis moved on from EMBO Reports, I have stepped in as the handling editor of your manuscript.

As you can see, the referees find that the study is significantly improved during revision and recommend publication. However, I need you to address the points below before I can accept the manuscript.

- Please provide 3-5 keywords for your study. These will be visible in the html version of the paper and on PubMed and will help increase the discoverability of your work.
- We note that the funding information is currently incomplete in our manuscript submission system - i.e. Predoctoral Fellowship from the Spanish Government (PRE2018-083178) and contract Investigo from the Generalitat Valenciana (INVEST/2022/203) are missing.
- We note that Appendix Table S2 is currently not called out in the text.
- The existing datasets need to be renamed as Dataset EV1 and Dataset EV2, and the callouts need to be updated accordingly.
- Please add page numbers to the Table of Contents of the Appendix file.
- Please resubmit Source Data as one zip file per figure.
- Please make the dataset PXD046448 publicly available and remove the reviewer access information from the manuscript text.
- Data Availability section is reserved for the primary datasets generated in the study. Therefore, please remove the following sentence from this section "All strains, plasmids and data are available from the authors upon request."
- Our production/data editors have asked you to clarify several points in the figure legends:
 - Please note that the figure legend style does not comply with the journal guidelines i.e. all the figure legends are in a run-on style.
 - Please note that a separate 'Data Information' section is required in the legends of all the figures.
 - Please note that figure title for figures EV1-5 are not provided.
 - Please indicate the statistical test used for data analysis in the legend of figure EV1d.
 - Please note that information related to n is missing in the legends of figures 1a-c, e; 2a-b; 3d; 5b, d, f, h; 7a-e; EV1a-c; EV2a; EV3a; EV5.
 - Please note that the error bars are not defined in the legends of figures EV1a-d; EV2a-c; EV3a-c; EV5.
 - Please note that the box plots need to be defined in terms of minima, maxima, centre, bounds of box and whiskers, and percentile in the legend of figure 5e.
- Papers published in EMBO Reports include a 'synopsis' and 'bullet points' to further enhance discoverability. Both are displayed on the html version of the paper and are freely accessible to all readers. The synopsis includes a short standfirst summarizing the study in 1 or 2 sentences (max 35 words) that summarize the paper and are provided by the authors and streamlined by the handling editor. I would therefore ask you to include your synopsis blurb and 3-5 bullet points listing the key experimental findings.
- In addition, please provide an image for the synopsis. This image should provide a rapid overview of the question addressed in the study but still needs to be kept fairly modest since the image size cannot exceed 550 (width) x 300-600 (height) pixels.

Thank you again for giving us to consider your manuscript for EMBO Reports, I look forward to your minor revision.

Kind regards,

Deniz Senyilmaz Tiebe

--

Deniz Senyilmaz Tiebe, PhD
Editor
EMBO Reports

Referee #1:

Upon revision, Ros-Carrero and co-workers provided a much improved version of the original manuscript. I acknowledge that the authors took the reviewers' comments very seriously and replied adequately. I genuinely think that the new experiments and improved data presentation strengthen the findings of the paper.

Referee #2:

In the revised version of the manuscript, the authors perform a series of valuable additional experiments to address the concerns raised by the reviewers. I think this has clearly improved the manuscript, and all my comments have been addressed. I think the manuscript is now suitable for publication, and I want to congratulate the authors on their interesting work.

Referee #3:

The authors have answered my points very well. I recommend it for publishing.

All editorial and formatting issues were resolved by the authors.

Dear Dr. Gomar-Alba,

Thank you for submitting your revised manuscript. I have now looked at everything and all is fine. Therefore, I am very pleased to accept your manuscript for publication in EMBO Reports.

Congratulations on a nice work!

Kind regards,

Deniz Senyilmaz Tiebe

--

Deniz Senyilmaz Tiebe, PhD

Editor

EMBO Reports

--
